# Superporous sponge prepared by secondary network compaction with enhanced permeability and mechanical properties for non-compressible hemostasis in pigs

Tianshen Jiang[1,2,9], Sirong Chen[1,2,9], Jingwen Xu[1,2,9], Yuxiao Zhang[1,2], Hao Fu[1,2], Qiangjun Ling[1,2], Yan Xu[3], Xiangyu Chu[4], Ruinan Wang[1,2], Liangcong Hu[4], Hao Li[5], Weitong Huang[1,2], Liming Bian [1,2,6,7] ✉, Pengchao Zhao [1,2,6,7] ✉ & Fuxin Wei[3,8] ✉

Developing superporous hemostatic sponges with simultaneously enhanced permeability and mechanical properties remains challenging but highly desirable to achieve rapid hemostasis for non-compressible hemorrhage. Typical approaches to improve the permeability of hemostatic sponges by increasing porosity sacrifice mechanical properties and yield limited pore interconnectivity, thereby undermining the hemostatic efficacy and subsequent tissue regeneration. Herein, we propose a temperature-assisted secondary network compaction strategy following the phase separation-induced primary compaction to fabricate the superporous chitosan sponge with highly-interconnected porous structure, enhanced blood absorption rate and capacity, and fatigue resistance. The superporous chitosan sponge exhibits rapid shape recovery after absorbing blood and maintains sufficient pressure on wounds to build a robust physical barrier to greatly improve hemostatic efficiency. Furthermore, the superporous chitosan sponge outperforms commercial gauze, gelatin sponges, and chitosan powder by enhancing hemostatic efficiency, cell infiltration, vascular regeneration, and in-situ tissue regeneration in non-compressible organ injury models, respectively. We believe the proposed secondary network compaction strategy provides a simple yet effective method to fabricate superporous hemostatic sponges for diverse clinical applications.

Non-compressible hemorrhage presents significant challenges in achieving hemostasis with conventional methods such as tourniquets or manual compression[1]. Failure to control non-compressible hemorrhage results in high mortality rates in both military and civilian contexts[2–4]. Thereby, development of various hemostatic materials capable of achieving rapid hemostasis in non-compressible perforation wounds[5–8] has received widespread attention, such as hemostatic sponges[9–11], adhesives[6,12–15] and powders[16,17]. Among these approaches, the shape-recoverable hemostatic sponges that restore initial shapes by absorbing blood to fill and then apply mechanical compression to the deep-narrow wound, have emerged as a promising strategy to control non-compressible hemorrhage[10,18–20]. Therefore, simultaneously enhancing liquid permeability and mechanical properties of sponges holds great promise in facilitating rapid shape recovery and

subsequently maintaining sufficient pressure on the wound, respectively, thereby building a robust physical barrier to greatly improve hemostatic efficiency.

Diverse strategies have been explored to prepare hemostatic sponges with porous structures, including direct lyophilization and utilization of foaming agents to increase liquid diffusion coefficient to accelerate shape recovery[21–25]. However, limited pore interconnectivity of these sponges results in restricted blood absorption, prolonged shape recovery time of several decades, and an inability to facilitate cell infiltration, tissue ingrowth, and vascularization[26–29], thereby compromising the hemostatic efficacy and subsequent tissue repair process. Moreover, increasing porosity and pore sizes to improve sponge permeability and rapid shape recovery typically compromises the sponge's mechanical properties[30], thereby undermining the sponge's physical structure and pressure maintenance on the wound. To our knowledge, developing hemostatic sponges with simultaneously enhanced permeability and mechanical properties remains challenging but highly desirable to achieve rapid and effective hemostasis for non-compressible hemorrhage.

Developing fabrication strategies to address the inherent limitations of key building block biomaterials, such as weak mechanical strength and poor bio-infiltration, has been the key objective of biomaterial research in the recent years. Compaction and densification of polymer networks to induce network alignment or form a dense entanglement[31,32] assisted by freeze-casting[33–35] or directional stretching[36–38] represent effective strategies to tune the mechanical properties of polymer structures with improved fatigue resistance and rapid shape recovery. Chitosan has emerged as a prominent material in the formulation of hemostatic agents, owing to its exceptional biocompatibility, biodegradability, anti-infection, and pro-coagulant properties. Increasing the pH of acidic chitosan solutions is known to trigger phase separation of chitosan, resulting in the formation of physically cross-linked crystals at the aggregated regions and subsequent formation of 3D chitosan networks[39,40]. However, phase separation-induced chitosan sponges typically possess low pore interconnectivity and limited pore sizes and therefore unsatisfactory hemostatic capability. To address this need for simultaneous enhancement in permeability and mechanical properties of chitosan sponges, we propose a simple temperature-assisted secondary network compaction (TA-2ndNC) strategy for fabricating the superporous chitosan sponge with highly-interconnected porous structure, rapid shape recovery capability, excellent fatigue resistance to mediate rapid non-compressible hemostasis and tissue regeneration.

Specifically, the superporous chitosan sponge (spCS) was fabricated by first cooling the phase-separated chitosan solution to the optimal pre-freeze drying temperature ($T_{pfd}$) of 0 °C. The optimal polymer chain mobility at the optimal $T_{pfd}$ enabled a controlled secondary polymer network reorganization and compaction during freeze drying to produce the spCS. Compared to the porous sponge (pCS) prepared without the secondary network compaction, spCS possesses highly-interconnected porous networks with large pore size and high porosity (269% higher than pCS), enhanced network density (53% higher than pCS), fatigue-resistant properties (retains 95% of the maximum stress after 100 cyclic compression at 85% strain), rapid and complete water-triggered shape recovery (0.84 s, 451% faster than pCS) and blood-triggered shape recovery (4.0 s, 410% faster than pCS), and demonstrated pro-coagulant properties. The alkylated superporous chitosan sponge (A-spCS) demonstrated good hemostatic capability in the non-compressible rat liver injury model (hemostasis in 13 s, 386% faster than commercial hemostat) and mini pig organ injury model (hemostasis in 39 s, 338% faster than commercial hemostat). The A-spCS retained in the injured rat livers can further promote cell migration, vascular regeneration, and guide tissue in-situ regeneration and integration, thereby demonstrating its promising potential for achieving hemostasis and subsequent in-situ tissue regeneration of non-compressible wounds.

## Results

### Fabrication of the superporous chitosan sponge (spCS) by temperature-assisted secondary network compaction (TA-2ndNC)

The superporous chitosan sponge (spCS) was prepared by the temperature-assisted secondary network compaction (TA-2ndNC) strategy as illustrated in Fig. 1a. The acidic chitosan solution was first treated with an alkaline solution to promote deprotonation and subsequent strengthening of hydrogen bonds to induce the gradual phase separation and therefore the primary compaction of chitosan. Then we present a simple temperature-assisted strategy to induce the secondary compaction of chitosan network by first cooling the phase-separated chitosan hydrogel to an optimal pre-freeze drying temperature ($T_{pfd}$) of 0 °C before starting freeze-drying. At a $T_{pfd}$ higher than 0 °C, i.e., 20 °C, the substantial polymer mobility results in the excessive volumetric expansion and drastic deformation of the chitosan sponge network during the subsequent free drying, which can compromise the integrity and toughness of overall sponge structure. At a $T_{pfd}$ lower than 0 °C, i.e., −80 °C obtained by flash freezing, the minimal chain mobility of fully frozen polymer severely restricts the secondary compaction of polymer network during freeze drying, thereby limiting the porosity and interconnectivity of the obtained sponge. In contrast, the optimal polymer chain mobility at 0 °C ensures a controlled secondary polymer network reorganization and compaction during freeze drying to produce the highly-interconnected superporous structure in chitosan sponge (Fig. 1a and Supplementary Fig. 1).

Micro-computed tomography (micro-CT), Scanning electron microscope (SEM), and porosity test (Fig. 2a, c and Supplementary Movie 1–5) results showed that spCS possessed highly-interconnected porous polymeric network with substantially large pore size/high porosity (1052.0 ± 208.90 μm, 88.42 ± 6.18%) (Fig. 2f, g). In contrast, the chitosan sponges obtained by freeze-drying the flash frozen acidic chitosan solution without primary compaction (CS, chitosan sponge) (17.80 ± 8.63 μm, 15.77 ± 3.03%) and by freeze-drying phase-separated chitosan hydrogel at a $T_{pfd}$ of −80 °C (pCS, porous chitosan sponge) (241.8 ± 96.63 μm, 23.95 ± 4.22%) possessed significantly smaller pore sizes and lower porosity compared with that of spCS (Fig. 2f, g). Furthermore, the density heat map of Micro-CT revealed that the average backbone density of spCS is significantly higher than that of CS, pCS (Fig. 2b, d and Supplementary Fig. 6), indicating that the secondary network reorganization under an optimal $T_{pfd}$ involves controlled localized condensation of the chitosan network. Although the chitosan sponge fabricated by freeze-drying phase-separated hydrogel at $T_{pfd}$ of 20 °C (epCS, excessively porous chitosan sponge) also possessed large pore size/high porosity (1648 ± 490.2 μm, 90.61 ± 2.84%), the excessive network dynamics caused by the lack of ice crystal formation led to drastic deformation of its network structure (Fig. 1a), significantly reducing the average network density (about 52% lower than spCS).

To further examine the effect of secondary network compaction on the structure of spCS, we utilized X-ray diffraction (XRD) to analyze the aggregate structure of the chitosan sponge. The (200/220) reflections of spCS shifted by about 0.3° toward a larger diffraction angle compared with that of pCS (Fig. 2e), indicating a decrease in the lattice spacing of the spCS[35,37] because of the rearrangement of hydrogen bonds between the chitosan chains during the secondary network compaction. Subsequently, we modified the surface of the spCS chitosan sponge with hydrophobic dodecyl chains to enhance blood coagulation ability (Fig. 2h and Supplementary Fig. 7). The results of X-ray photoelectron spectroscopy (XPS) showed that the dodecanal modification rate of alkylated spCS (A-spCS) was 27.20 ± 3.48% (Fig. 2i). The interconnected structure and pore diameters/porosity (993.8 ± 159.9 μm, 88.02 ± 7.1%) of the A-spCS were similar to spCS, suggesting that no significant structural impact of alkylation modification. Moreover, our laboratory-scale synthesis

yielded ~50 ml of spCS per batch in a 250 mL beaker and showed stable batch-to-batch reproducibility, indicating the promising potential for large-scale production of spCS (Supplementary Fig. 2).

These findings demonstrate the superiority of temperature-assisted secondary network compaction strategy to enhance both the liquid permeability and mechanical properties of the hemostatic sponge simultaneously (Fig. 1b). In contrast, despite prior reports on the effects of freeze-drying temperature on chitosan sponge structures, the temperature-assisted secondary network compaction (TA-2nd NC) strategy diverges markedly from the prior methods, in terms of procedure, principles and outcomes (Supplementary Fig. 1). Conventional chitosan sponges are typically prepared by directly freeze-drying chitosan solutions at −20 °C, −80 °C, and −196 °C by using ice crystals as pore templates, and the obtained sponges generally exhibit

small pore size (10 to 250 μm), low porosity (23% to 80%), limited water absorption (5.5−9.5 g/g in 24 h), and reduced maximum stress upon compression[41–44].

## A-spCS possesses enhanced blood absorption capacity and mechanical properties

Ideal hemostatic sponges with high liquid permeability can realize rapid blood-triggered shape recovery and subsequently establish a physical hemostatic barrier to achieve high hemostatic efficacy. Therefore, rapid shape recovery after compression is essential for hemostasis sponges. We first evaluated the microstructures of compressed sponges and their shape recovery after absorbing water or blood by SEM (Fig. 3a, b). Superporous hemostatic sponges (spCS/A-spCS) restored their circular pore structures after absorption of water or blood. Furthermore, spCS

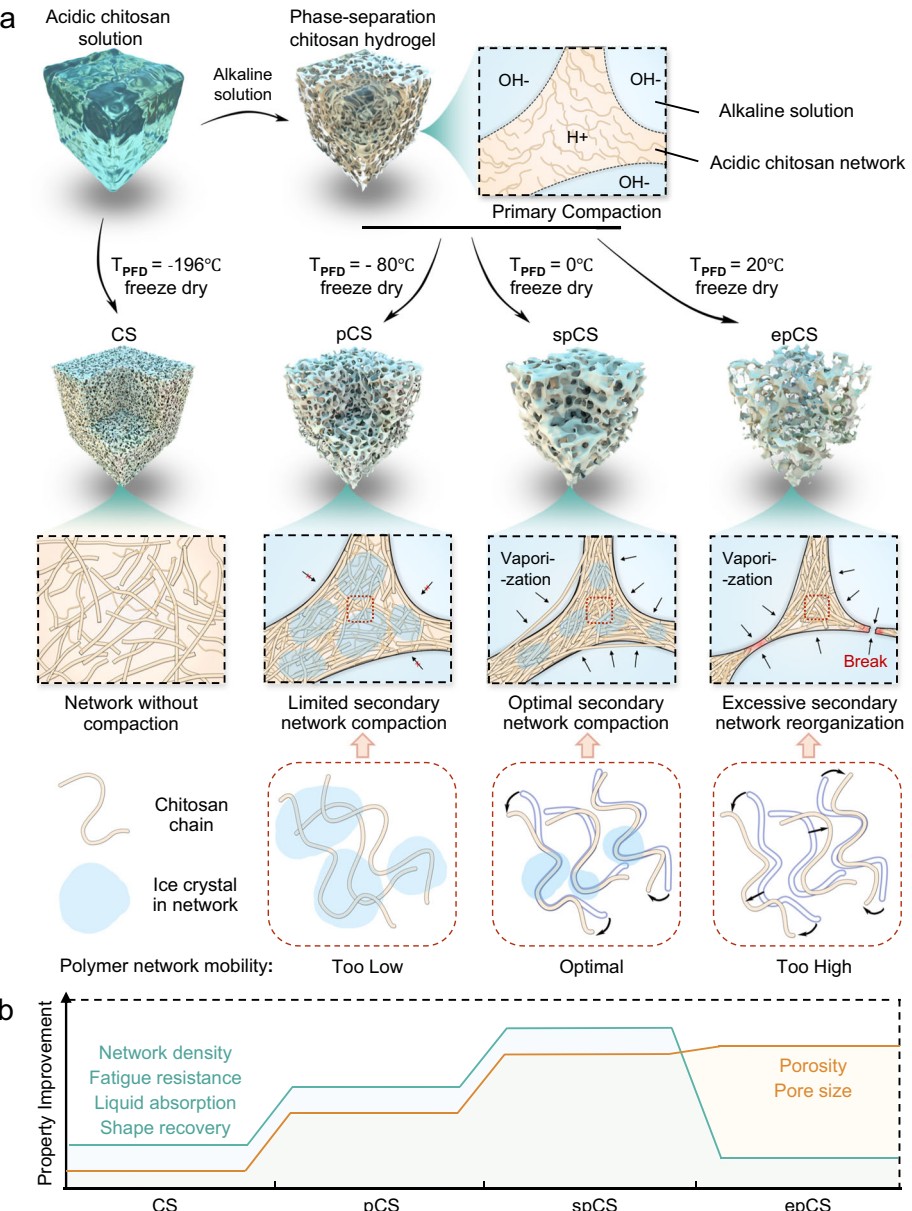

**Fig. 1 | Fabrication of the superporous chitosan sponge (spCS) by temperature-assisted secondary network compaction (TA-2ndNC). a** Fabrication of sponges by the TA-2ndNC strategy. The chitosan solution was first treated with phase separation and then was subjected to pre-freezing treatments at −80 °C, 0 °C, and 20 °C, followed by freeze drying to fabricate porous chitosan sponge (pCS), spCS, and excessively porous chitosan sponge (epCS), respectively. The chitosan sponge (CS) was obtained by freeze-drying the flash-frozen acidic chitosan solution without network compaction. **b** The TA-2ndNC strategy was employed to fabricate the sponges with tunable network density, fatigue resistance, liquid absorption capacity, shape recovery ability, pore size, and porosity through the modulation of secondary network reorganization.

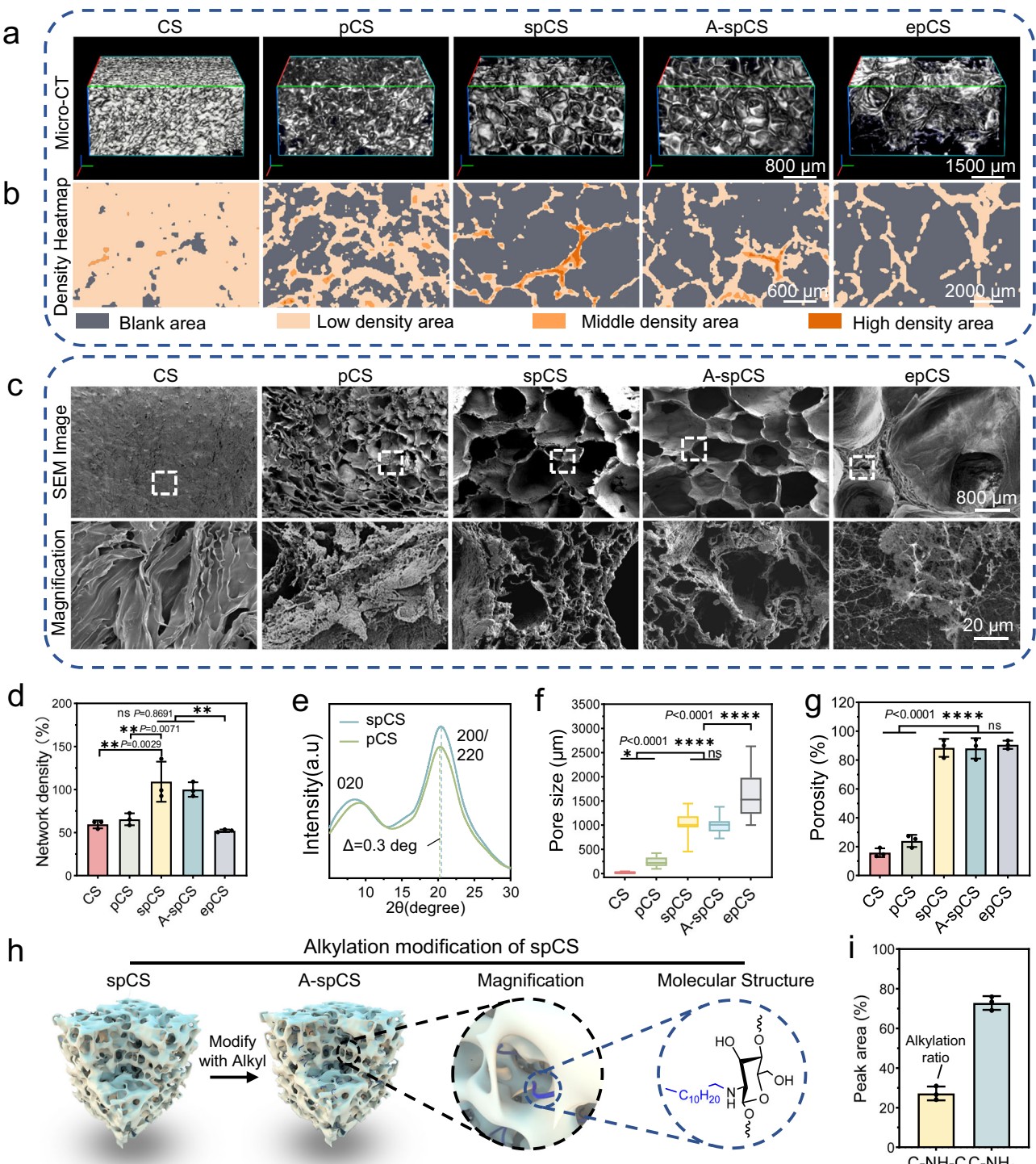

**Fig. 2 | Structural and chemical characterization of superporous chitosan sponge (spCS). a** Micro-CT images of the microstructure of chitosan sponge (CS), porous chitosan (pCS), spCS, alkylated-superporous chitosan sponge (A-spCS), and excessively porous chitosan sponge (epCS). **b** Density heatmaps of the chitosan networks of CS, pCS, spCS, A-spCS, and epCS. **c** Scanning electron microscope (SEM) images of the microstructure of CS, pCS, spCS, A-spCS, and epCS. **d** Average network density of CS, pCS, spCS, A-spCS, and epCS. **e** X-ray diffraction (XRD) spectrum shows the chitosan lattice structure of pCS and spCS. **f** Pore diameter of CS, pCS, spCS, A-spCS, and epCS ($n = 25$, each data point is randomly from 3 independent samples). Lower whisker represents for end smallest value, upper whisker represents for largest value, line inside the box represents for median value, bounds of the box represents for first quartile and third quartile. **g** Porosity of CS, pCS, spCS, A-spCS and epCS. **h** The schematic diagram of alkyl chain modification for spCS. **i** The peak area ratio of the C-NH$_2$ and C-NH-C peaks in the N1$s$ peak of X-ray photoelectron spectroscopy (XPS) spectra of A-spCS. Values and error bars in (**d**, **g**, **i**) represent the mean and standard deviation ($n = 3$ independent samples) $P > 0.05$ (ns), **$P < 0.01$, ***$P < 0.001$, ****$P < 0.0001$ (one-way ANOVA to compare multiple groups or two groups).

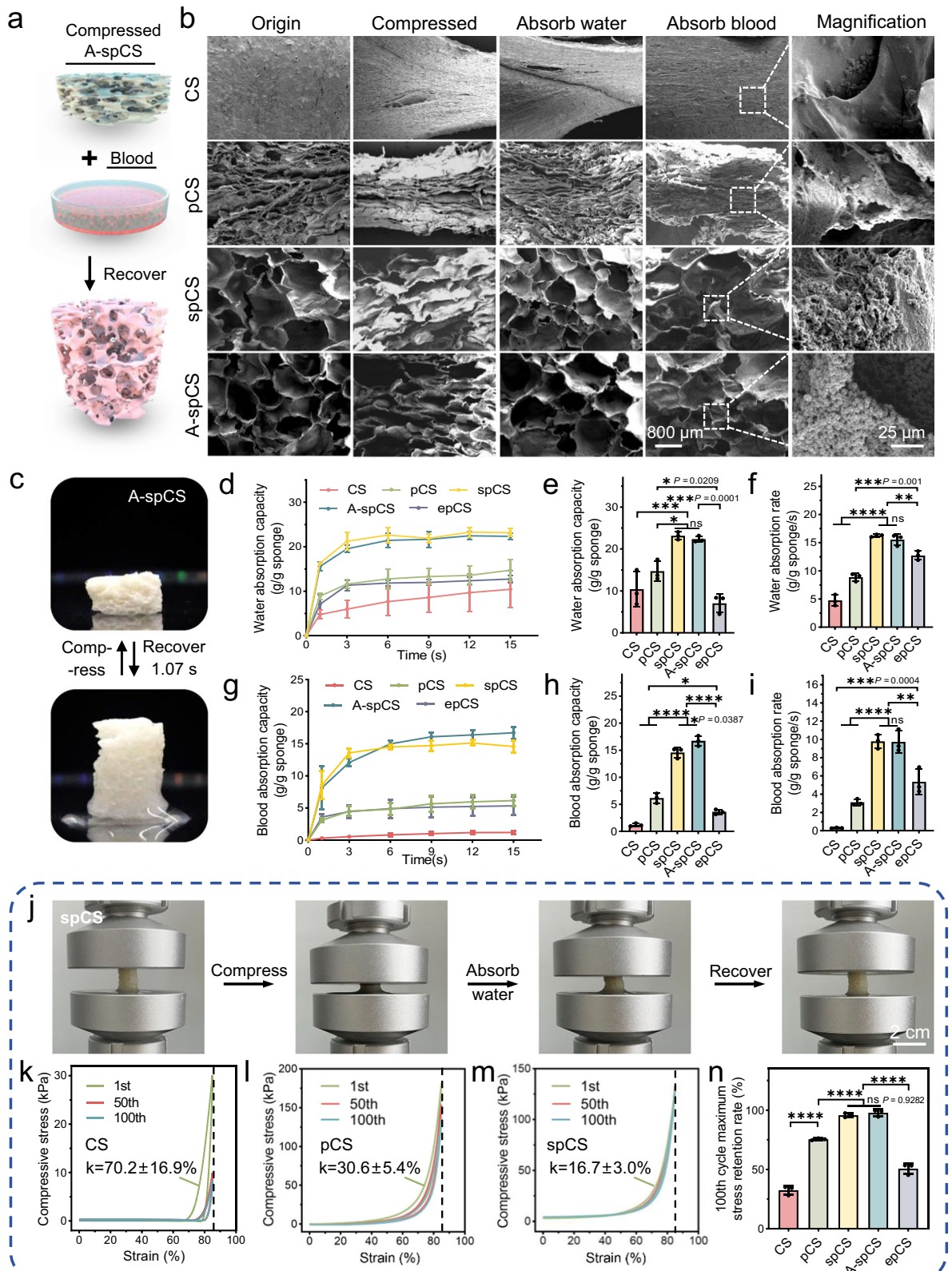

**Fig. 3 | Evaluation of liquid permeability and mechanical properties of the superporous chitosan sponge (spCS).** **a** Schematic diagram of shape recovery tests of compressed alkylated-superporous chitosan sponge (A-spCS) after absorbing blood. **b** SEM images showed the microstructures of the compressed sponges after compression and water/blood absorption. **c** Rapid shape recovery of the A-spCS sponge after absorbing water. **d**, **e**, **f** Water absorption capacity of chitosan sponge (CS), porous chitosan sponge (pCS), spCS, A-spCS and excessively porous chitosan sponge (epCS). **g**, **h**, **i** Blood absorption capacity of CS,

pCS, spCS, A-spCS and epCS. **j–m** Stress-strain cyclic curves of CS, pCS, and spCS (k in the figures represents the hysteresis in the first cycle of each group). **n** Maximum stress retention rates of CS, pCS, spCS, A-spCS, and epCS after 100 times stress-strain cycles.Values and error bars in (**e**, **f**, **g–i**, **n**) represent the mean and standard deviation (*n* = 3 independent samples), *P* > 0.05 (ns), **\*\**P* < 0.01, **\*\*\**P* < 0.001, **\*\*\*\**P* < 0.0001 (one-way ANOVA to compare multiple groups or two groups).

(0.84 ± 0.13 s, 4.0 ± 1.13 s) and A-spCS (1.07 ± 0.30 s, 4.37 ± 1.36 s) can achieve complete shape recovery after absorbing water/blood (Supplementary Figs. 9, 10 and Supplementary Movie 8, 12), respectively, which can be attributed to the highly-interconnected superporous network induced by temperature-assisted secondary network compaction. In contrast, SEM results together with the water-triggered and blood-triggered shape recovery demonstrated that CS and pCS cannot recover to their original shape after absorbing blood due to their limited interconnected structures and small pore size (Supplementary Movie 6, 7, 10, 11). Specifically, pCS can only reach a 70.8% recovery ratio and no significant shape recovery occurred in CS due to its compacted structure within 5 minutes after absorbing blood (Supplementary Fig. 10). Compressed epCS was entirely unable to recover to its original shape after absorbing water or blood (Supplementary Fig. 11, 12 and Supplementary Movie 9, 13). Despite that epCS possesses an interconnected large-pore network structure, the low network density renders it incapable of withstanding pressure and undergoes irreversible structural damage. To further evaluate the water and blood absorption of the A-spCS sponges, we analyzed the dynamic curves of water and blood absorption over time (Fig. 2d-i). Compared with CS, pCS and epCS, the maximum water/blood absorption capacity and absorption rate of spCS and A-spCS were significantly higher. Conversely, CS and pCS impede the penetration of water and high-viscosity blood as their limited porosity and pore size. It is interesting to note that, due to the interconnected pore structure of epCS, compressed epCS exhibits a higher liquid absorption rate compared to CS and pCS. However, the liquid absorption capacity of epCS is significantly lower than that of pCS, which can be attributed to the damage of network structure after compression and the resulting decrease in final porosity as the lack of shape recovery capability. This underscores that both a highly-interconnected porous structure and good fatigue resistance for shape recovery are indispensable for achieving rapid liquid absorption and further efficient hemostasis. To further demonstrate that 0 °C is the optimal $T_{pfd}$, we tested the shape recovery capability of sponges prepared by setting $T_{pfd}$ at -40, -20, -5, and 5 °C, respectively. The results indicate that as $T_{pfd}$ approached 0 °C, the absorption capacity increased but remained lower compared with that of spCS (Supplementary Fig. 8), and this finding indicates that the optimal secondary network reorganization achieved at 0 °C $T_{pfd}$ can enhance the toughness, shape recovery, and liquid absorption capacity of spCS. Therefore, these results suggest that the spCS and A-spCS are suitable for rapid hemostasis applications due to their rapid shape recovery ability and liquid absorption ability.

The mechanical strength and fatigue-resistant ability of a hemostatic sponge after shape recovery are the other important factors in providing a stable and strong physical hemostatic barrier to apply pressure on surrounding tissues. The spCS and A-spCS sponges were tested for 100 cycles of stress-strain tests to evaluate their fatigue resistance ability (Fig. 3j). Sponges were compressed to 85% strains for every cycle. Results showed that spCS and A-spCS still retained over 95% of their maximum stress, indicating the fatigue-resistance ability of the sponges with optimal network reorganization (Fig. 3m, n and Supplementary Fig. 17). While CS, pCS and epCS can only retain about 33%, 75% and 51% of maximum stress, respectively, indicating that they were more likely to suffer from structural damages under the same compression conditions (Fig. 3k, l, n, and Supplementary Fig. 18). Meanwhile, after absorbing liquid under 85% compression and shape fixation state, spCS can still sustain compressive pressures of 123.72/32.76 kPa, significantly higher than those of epCS (34.36, $P = 0.0003$, 95% CI = 46.71 to 132.0/14.64 kPa, $P = 0.0005$, 95% CI = 9.074 to 27.17) and commercially available gelatin sponge (21.31, $P < 0.0001$, 95% CI = 59.76 to 145.0/0.184 kPa, $P < 0.0001$, 95% CI = 23.53 to 41.62) (Supplementary Figs. 13, 20). Although epCS exhibited favorable porosity and high interconnectivity with a large pore structure, its compromised mechanical properties will undermine the capacity to sustain sufficient pressure to wounds. This finding further underscores the importance of combining good mechanical performance and rapid liquid permeability in the chitosan sponges for effective hemostasis. Moreover, larger hysteresis loops were observed from the stress-strain curves of the CS, pCS, and epCS than spCS and A-spCS (Supplementary Fig. 19), demonstrating large dissipation energy and limited resistance to the compression cycles[45,46]. Therefore, the superior mechanical performance such as fatigue-resistant ability and rapid shape recovery can be ascribed to the network reorganization of chitosan sponges with TA-2ndNC strategy under optimal $T_{pfd}$, in which highly-interconnected superporous structures contribute to rapid shape recovery, while the rearrangement of physical cross-linking enhances the fatigue-resistant ability of network structure (Fig. 1).

## A-spCS exhibits good pro-coagulant abilities in vitro

The adhesion and aggregation of RBCs and platelets play a key role in blood clotting and hemostasis, which are also associated with platelet activation[9,47,48]. We first employed the Blood Clotting Index (BCI) test to quantitatively evaluate the pro-coagulant ability of spCS and A-spCS sponges, in which a lower BCI value indicates superior pro-coagulant properties (Fig. 4c). CS, pCS and commercially available gauze, gelatin sponge (GS), CELOX, CELOX-E, were set as controls. CELOX and CELOX-E are FDA-approved hemostatic chitosan powder and chitosan-based gauze for medical emergencies, respectively. A-spCS sponges exhibited the lowest BCI values, indicating the superior and effective pro-coagulant performance of the A-spCS. We further evaluated the adhesion of RBCs and platelets on different hemostatic materials, which showed that A-spCS had the highest number of adhering RBCs and platelets than gauze, GS, CELOX, CELOX-E, CS, and pCS, while the adhesion abilities of RBCs and platelets on the spCS are second only to A-spCS group and better than other control groups (Fig. 4d, e). Consistently, SEM images confirmed that spCS and A-spCS exhibited the densest RBCs aggregation, compared with other controls (Fig. 4b). Furthermore, to evaluate the efficacy of chitosan sponges to promote coagulation in the situation of non-compressible wound bleeding with higher-volumes blood loss, we conducted BCI test, blood cell adhesion test, and platelet adhesion tests by using three times the normal blood volume (Supplementary Fig. 21). The results indicate that due to the higher blood absorption rate and volume of spCS, it demonstrated superior in vitro procoagulant ability compared to pCS. Therefore, we believe that the highly-interconnected superporous structures of spCS/A-spCS can promote the rapid absorption of blood and concentrate blood cells, which facilitate platelets activation and blood coagulation[49,50]. Moreover, alkylated chitosan is known to enhance RBCs and platelets adhesion via electrostatic interaction and alkylation[10,13,49], further inducing platelets activation and promoting blood clotting. Therefore, the pro-coagulant ability of A-spCS can be attributed to the synergistic effect of highly-interconnected superporous structures, electrostatic interaction of chitosan, and modification of alkyl chains.

## A-spCS is anti-bacterial biocompatible

In addition to promoting hemostasis, preventing infection is crucial in the management of bleeding and may ultimately determine a patient's prognosis. We examined the antibacterial efficacy of spCS and A-spCS, in comparison to other commonly used hemostats against three bacteria strains, *Staphylococcus aureus* (*S. aureus*), *Escherichia coli* (*E. coli*) and *Pseudomonas aeruginosa* (*P. aeruginosa*) (Fig. 5a–c and Supplementary Figs. 22, 23), which are the most representative pathogens associated with wound infections[51,52]. Through quantitative analysis (Fig. 5d, e), our findings demonstrated that A-spCS exhibited one of the lowest colony-forming unit (CFU) count of *S. aureus, E. coli,* and *P. aeruginosa* after co-culturing for 2 h. In contrast, the spCS group demonstrated similar antibacterial capability with CS and pCS groups, but much better than gauze and gelatin sponge (GS) groups, thus indicating that the

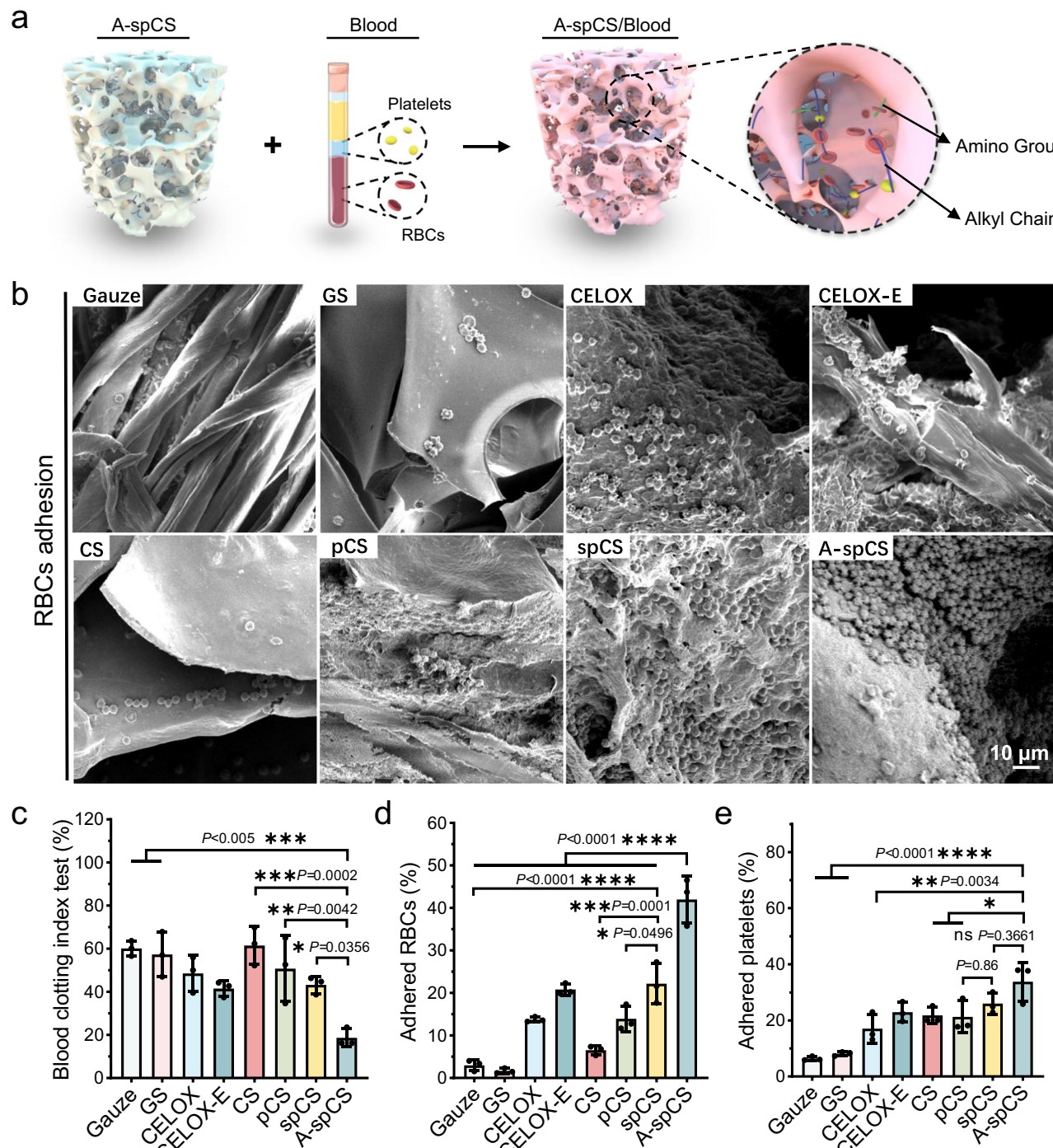

**Fig. 4 | Coagulation effect of various hemostatic materials and blood cell adhesion ability. a** Schematic diagram demonstrating the pro-coagulant mechanism of the alkylated-spCS (A-spCS). **b** SEM images showing the adhesion of red blood cells on gauze, gelatin sponge (GS), CELOX, CELOX-E, chitosan sponge (CS), porous chitosan sponge (pCS), superporous chitosan sponge (spCS), and A-spCS. **c** 10 min Blood Clotting Index (BCI) values of hemostatic materials. **d, e** Percentage of red blood cells and platelets adhered on different hemostatic materials. Values and error bars in (**c**–**e**) represent the mean and standard deviation (*n* = 3 independent samples), *P* > 0.05 (ns), **\*\**P* < 0.01, **\*\*\**P* < 0.001, **\*\*\*\**P* < 0.0001 (one-way ANOVA to compare multiple groups or two groups).

positively charged amino groups of chitosan, rather than the highly-interconnected superporous structure induced by TA-2ndNC, are more conducive to enhancing the antibacterial ability. Furthermore, A-spCS demonstrated antibacterial efficacy against the three bacteria strains after co-incubation for 4 and 6 h (Supplementary Fig. 24), and penicillin G-loaded A-spCS sustained prolonged antibacterial effects for up to 2 days (Supplementary Fig. 25). Therefore, the potent antibacterial properties of A-spCS can be notably attributed to the interaction between the

positively charged amino groups[53] and the hydrophobic alkyl chains[54] with the negatively charged bacterial membrane, which disrupts bacterial membrane and subsequent bacterial lethality.

To further assess the biocompatibility of A-spCS, we conducted cell survival testing to confirm that the A-spCS sponges demonstrated good cytocompatibility when demonstrating its perfect efficacy in antibacterial properties. Specifically, we performed fluorescent live/ death staining of 3T3 fibroblasts and LX-2 hepatic stellate cell as well as

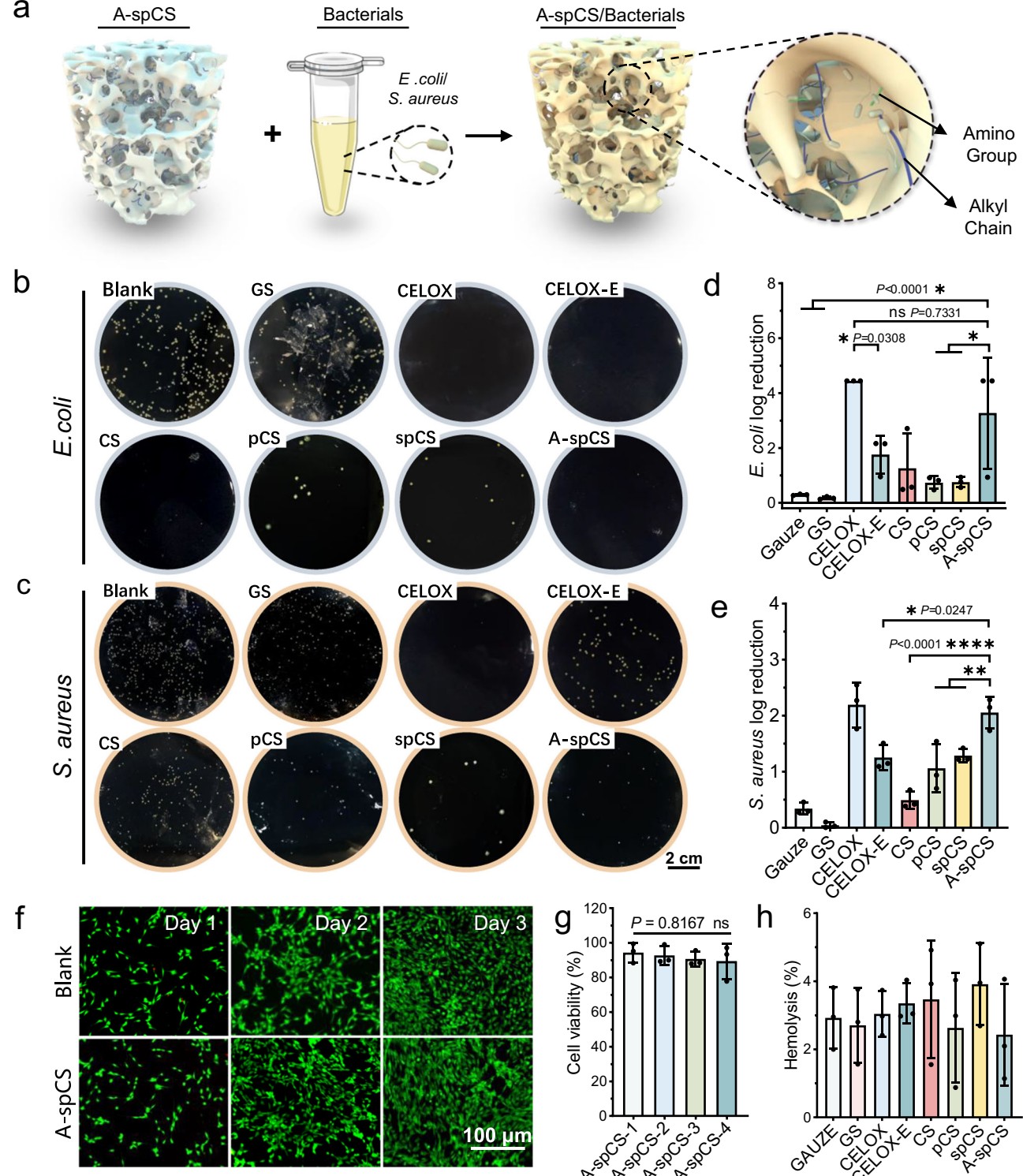

**Fig. 5 | In vitro anti-infective properties of various hemostatic materials.**
**a** Schematic showing the antibacterial mechanism of alkylated-superporous chitosan sponge (A-spCS). **b, c** Photographs of colonies of *E. coli.* and *S. aureus* grown on LB agar plates after contact with blank, gauze, GS, CELOX, CELOX-E, chitosan sponge (CS), porous chitosan sponge (pCS), superporous chitosan sponge (spCS), and A-spCS. **d, e** Corresponding statistical results of log reduction of *E. coli.* and *S. aureus.* **f** Fluorescence microscopy images of live-dead staining of 3T3 fibroblasts after 1, 2, and 3 days of culture in A-spCS

extract (*n* = 3). **g** The MTT tests revealed the cell viability of 3T3 fibroblasts cultured in different mass fractions (10, 20, 30, 40 mg/mL) of A-spCS extracts. **h** Hemolysis tests of Gauze, GS, CELOX, CELOX-E, CS, pCS, spCS and A-spCS. Values and error bars in (**d, e, g, h**) represent the mean and standard deviation (*n* = 3 independent samples), *P* > 0.05 (ns), **_P_ < 0.01, ***_P_ < 0.001, ****_P_ < 0.0001 (one-way ANOVA to compare multiple groups or two groups).

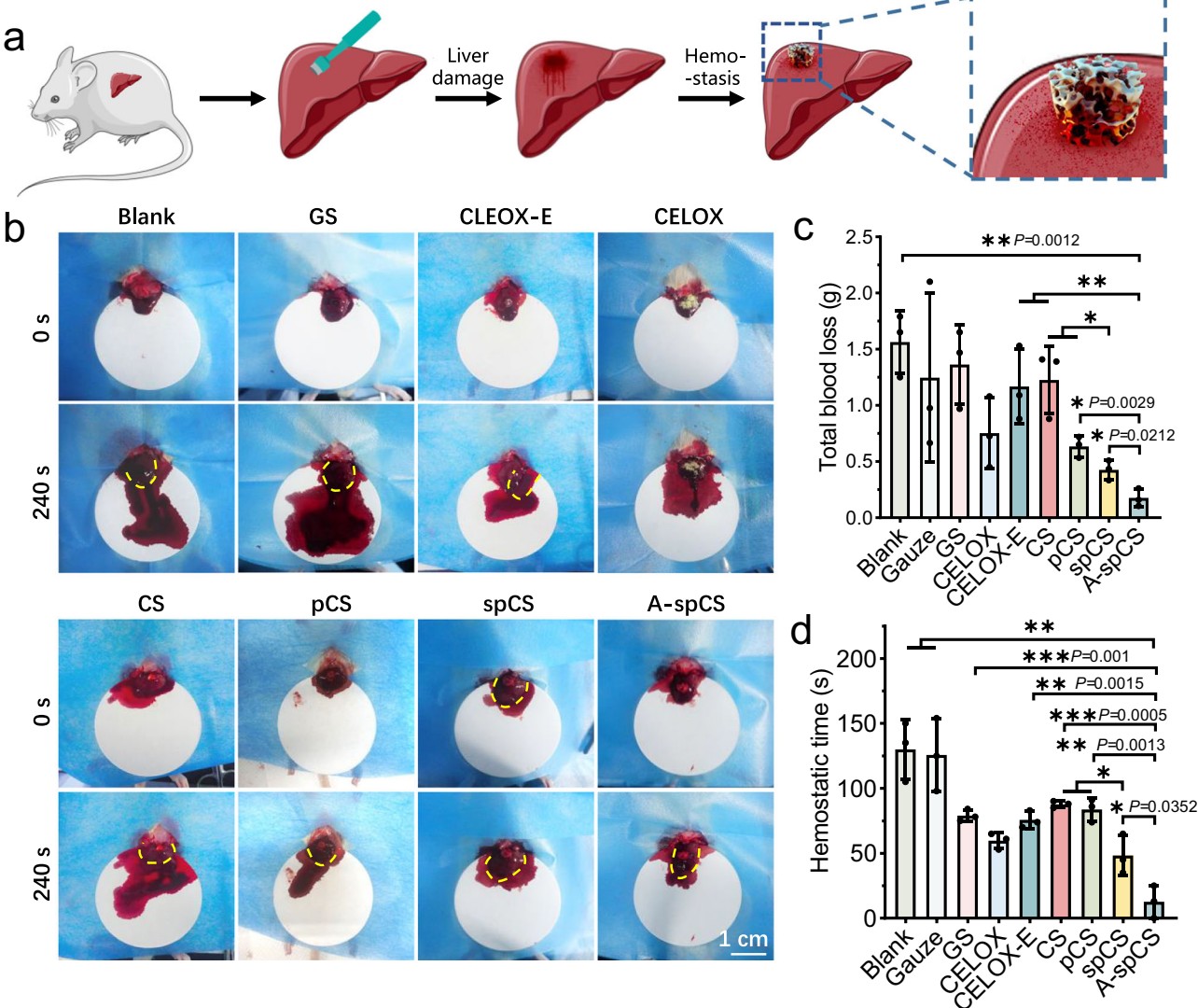

**Fig. 6 | In vivo hemostatic effect of alkylated-superporous chitosan sponge (A-spCS) in SD rats. a** Schematic diagram of the rat liver perforation model and the hemostasis process. **b** Photographs of hemostatic effects of blank, gauze, gelatin sponge (GS), CELOX, CELOX-E, chitosan sponge (CS), porous chitosan sponge (pCS), superporous chitosan sponge (spCS), and A-spCS. The yellow dotted line is the boundary of the liver. **c, d** Total blood loss and hemostasis time of blank, gauze, GS, CELOX, CELOX-E, CS, pCS spCS, and A-spCS groups. Values and error bars in (**c, d**) represent the mean and standard deviation ($n$ = 3 independent samples), $P > 0.05$ (ns), **$P < 0.01$, ***$P < 0.001$, ****$P < 0.0001$ (Two-tailed Student's $t$ test to compare two groups).

the MTT/CCK-8 tests to evaluate the effects of co-culturing with A-spCS leachate in different mass fractions (10, 20, 30, and 40 mg/mL) for 1, 2 and 3 days. Our statistical analysis of the resulting data revealed that A-spCS did not reduce the survival rate of 3T3 cells and LX-2 cells (Fig. 5f, g and Supplementary Fig. 26). We further conducted hemolysis tests to evaluate the hemocompatibility of A-spCS, revealing no significant difference between A-spCS and other groups. None of the groups exceeded the upper limit standard of 5% (Fig. 5h and Supplementary Fig. 27) according to regulations formulated by the American Society for Testing and Materials (ASTM)[55], suggesting that A-spCS possesses favorable biocompatibility, which represents a promising candidate for future applications in hemostasis settings.

### A-spCS mediates rapid and enhanced hemostasis in a large animal model

The hemostatic potential of spCS and A-spCS was first evaluated in a Sprague-Dawley (SD) rat liver perforation wound model (Fig. 6). Notably, the area of bloodstain on the filter paper in the A-spCS group was smallest compared to the gauze, GS, CELOX, CELOX-E, CS, pCS and

spCS groups (Fig. 6b, Supplementary Fig. 28, and Supplementary Movie 11). Our statistical analysis further revealed that the total blood loss in the A-spCS group was markedly lower than that in the other groups (Fig. 6c), and the hemostasis time was significantly shorter (Fig. 6d). Besides, the hemostatic time of spCS was also remarkably shorter than that of the pCS, CS group, emphasizing that the highly-interconnected superporous structure induced by secondary network reorganization achieves rapid shape recovery and constructs a robust hemostatic physical barrier.

To further evaluate the hemostatic efficacy of A-spCS in vivo, we next employed a penetrating wound model with a diameter of 18 mm on spleens and livers of Bama pigs (Fig. 7a and Supplementary Movie 16). The untreated wound continued to bleed for over four minutes, whereas A-spCS effectively stopped the bleeding within only 30 s in liver hemostasis (Fig. 7d, f). Notably, A-spCS demonstrated superior hemostatic capacity in pig spleen wounds, compared to the groups treated with commercially powder hemostat (CELOX) (13.43 ± 7.79 g vs. 44.435 ± 4.95 g, $P = 0.0301$, 95% CI = 3.388 to 52.93) (Fig. 7c, e and Supplementary Movie 15). Similarly, A-spCS displayed

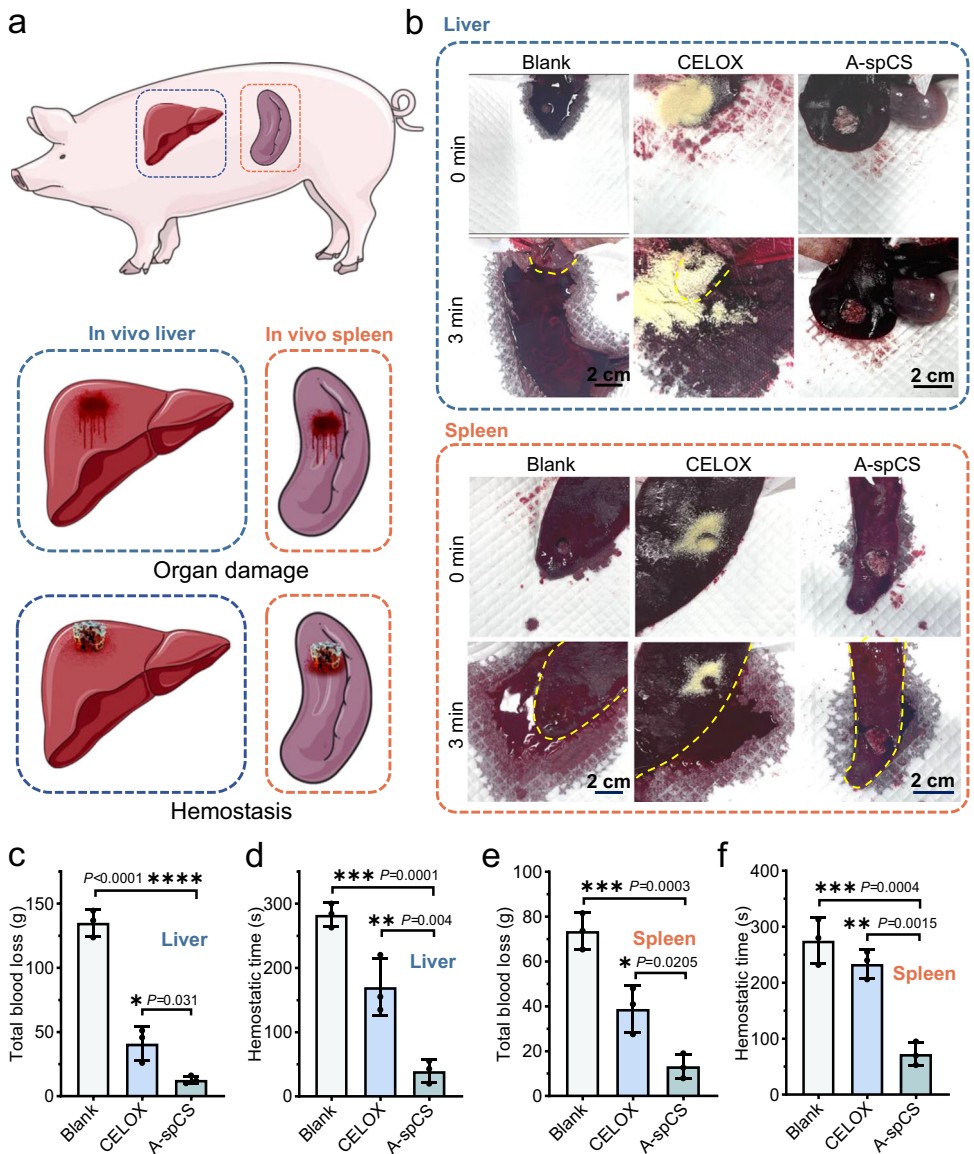

**Fig. 7 | In vivo hemostatic effect of alkylated-superporous chitosan sponge (A-spCS) in Bama minipigs. a** Schematic diagram of the hemostatic process of A-spCS acting on the pig liver and spleen perforation model. **b** Photos of liver/spleen hemostatic effect in blank, CELOX, and A-spCS groups. **c, e** Liver/spleen total blood loss in blank, CELOX, and A-spCS groups. **d, f** Liver/spleen hemostasis times of blank, CELOX, and A-spCS groups. Values and error bars in (**c–f**) represent the mean and standard deviation ($n$ = 3 independent samples), $P > 0.05$ (ns), **$P < 0.01$, ***$P < 0.001$, ****$P < 0.0001$ (one-way ANOVA to compare multiple groups or two groups).

remarkable efficacy in controlling bleeding in pig liver wounds when compared with CELOX powder (Fig. 7b, c, d). Those findings can be attributed to the highly-interconnected superporous network structure, the robust mechanical properties, and alkylation of A-spCS that facilitates rapid blood absorption, thereby rapidly restoring its shape, exerting enough pressure on the wound surface, and enhancing pro-coagulant efficacy, and eventually triggering a cascade of hemostasis and forming blood clots.

## A-spCS promotes in situ liver regeneration in an animal liver injury model

Hemostatic sponges that can guide tissue growth in situ allow the sponges to stay within the wound, thereby reducing the possibility of secondary bleeding when removing hemostats. We evaluated the in situ pro-regeneration ability of CELOX, CS, pCS, and A-spCS in a SD rat liver regeneration model (Fig. 8a). The H&E staining on the sections of rat liver tissues harvested at four weeks after the hemostasis

revealed that significantly more cell numbers and tissue ingrowth area in the A-spCS group compared with the CELOX, CS and pCS groups (Fig. 8b, c, d). We further evaluated capillary generation with von Willebrand factor (VWF) (Fig. 8b), a factor formed in the vessels. A-spCS contained a much larger capillary density than CELOX and CS groups (Fig. 8e). Moreover, the ingrowth of liver parenchymal cells (LPC) and tissue regeneration was confirmed by albumin (ALB)-positive cells observed in A-spCS. Liver development was also evaluated by immunostaining against hepatocyte nuclear factor-4α (HNF-4α), a transcription factor associated with the generation of hepatocyte, and the results indicated significantly higher number of liver cells in the A-spCS group than the CELOX, CS, and pCS group (Fig. 8f). The improvement of tissue ingrowth and vascularization of the A-spCS can be ascribed to highly-interconnected superporous network structure of A-spCS with large pore size/high porosity.

Additionally, the complete recovery of the liver in healthy individuals typically takes weeks to months following trauma or surgery[56].

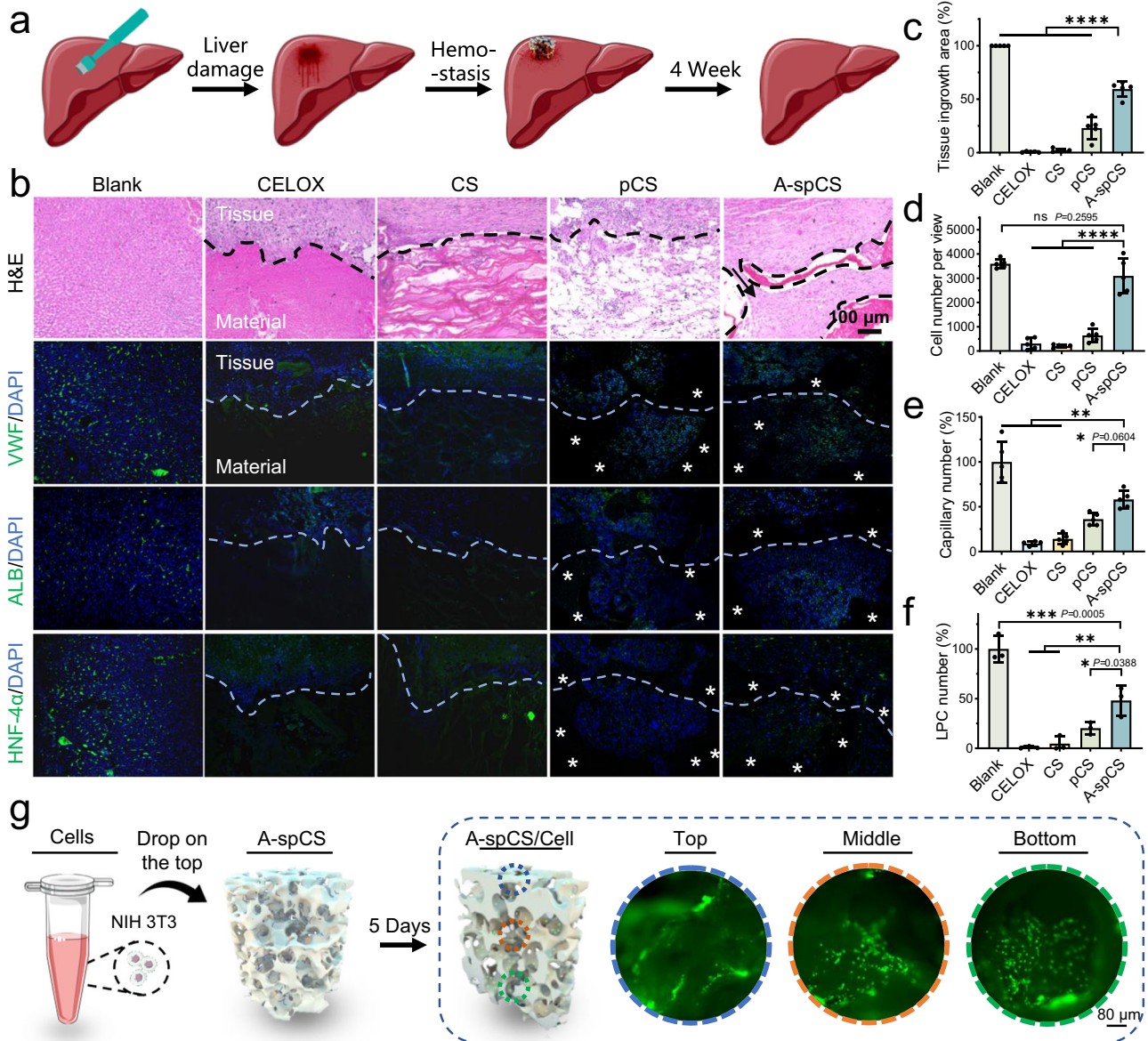

**Fig. 8 | Liver regeneration in rat models after implantation of the CELOX, chitosan sponge (CS), porous chitosan sponge (pCS), and alkylated-superporous chitosan sponge (A-spCS). a** Schematic diagram of the process of in situ liver regeneration experiment. **b** H&E staining showing tissue ingrowth. DAPI staining showed that the host cells migrated into the interior of the sponge. Immunofluorescent staining for von Willebrand factor (VWF), albumin (ALB)-positive cells, and hepatocyte nuclear factor-4α (HNF-4α) revealed the obvious infiltration of capillaries and liver parenchymal cells (LPC) within the A-spCS sponges, respectively. The dashed line represents the interface between material and tissue. * represents the implanted material. **c**–**f** Quantification of tissue in growth area, cell number, capillary number per view, and LPC number per view within the CELOX, CS, pCS, and A-spCS. **g** Schematic diagram of in vitro cell migration experiments within A-spCS sponges. Values and error bars in (**c**–**e**) represent the mean and standard deviation ($n = 5$ independent samples). Values and error bars in (**f**) represent the mean and standard deviation ($n = 3$ independent samples), $P > 0.05$ (ns), **$P < 0.01$, ***$P < 0.001$, ****$P < 0.0001$ (one-way ANOVA to compare multiple groups or two groups).

Previous studies have reported that certain liver scaffolds, due to the mechanical fragility and rapid degradation, can lead to the premature loss of the physical and biochemical microenvironment for encapsulated cells[57]. Compared with gelatin sponges, which degrade almost entirely within two weeks, A-spCS achieved partial in vivo degradation at four weeks consistent with prior reports[58] and can therefore serve as a protective scaffold to support cell migration, which is important to vascular development and tissue repair (Supplementary Fig. 30).

To further confirm the cell infiltration capability of A-spCS, we also performed the cell migration tests in vitro (Fig. 8g). Specifically, we dropped the NIH 3T3 fibroblast suspension on the upper surface of

the cylindrical A-spCS sponges. After 5 days of co-culture, the A-spCS sponges were cut vertically, and the cells were stained with Calcein AM, a cell-permeant dye that can be used to determine cell viability. A confocal laser scanning microscope was used to observe the number and morphology of cells on the top, middle, and bottom of A-spCS. Fluorescent images revealed that cells can penetrate into the sponge and grow robustly even in the middle and bottom of A-spCS. Furthermore, to better simulate the actual conditions in the liver, we selected LX-2 hepatic stellate cells, which play a crucial role in liver injury repair[59,60], for further validation of cell migration. Compared with the control group of CS and pCS, spCS promoted the growth and spreading of LX-2 cells (Supplementary Fig. 31).

In this study, we present a hemostatic sponge prepared by the temperature-assisted secondary network compaction strategy for the treatment of non-compressible hemorrhage and subsequent guidance of tissue regeneration. The resulting A-spCS sponges possesses highly-interconnected superporous structure, robust mechanical properties, procoagulant properties, good biocompatibility, and antimicrobial performance, enabling rapid absorption of large quantities of blood and complete shape recovery, thereby achieving efficient hemostasis in non-compressible SD rat and pig organs injury models, which outperforms commercial gauze, gelatin sponges, chitosan powder, and chitosan gauze in terms of hemostatic efficacy. Furthermore, the retained A-spCS in the body can further facilitate cell migration, vascular regeneration, and guide tissue in-situ regeneration and integration, thereby reducing the possibility of secondary bleeding when removing hemostats.

Overall, the ability of A-spCS sponges to quickly stop bleeding and in situ guide tissue repair makes it possess the potential of an emergency hemostatic for clinical application in solving non-compressible wounds.

## Methods

### Protocol approval
The experimental operations in this study strictly complied in accordance with the appropriate ethical guidelines, and the use of animals in this study was approved by the South China University of Technology Animal Care and Use Committee (Permit Number 2022032)

### Materials
Chitosan (degree of deacetylation: 95%), dodecanal, sodium triacetoxyborohydride, sodium bicarbonate, sodium dihydrogen phosphate, disodium hydrogen phosphate, absolute, and dichloromethane were purchased from J&K Scientific (Beijing China). 4% paraformaldehyde was purchased from White Shark Biotechnology. Triton-X-100 was purchased from Merck Life Sciences.

### Preparation process of A-spCS
Firstly, 2.5 wt% chitosan was dissolved in 1.25 wt% acetic acid solution. After stirring, the solution was placed in a refrigerator at 4 °C for 48 h. The homogenized chitosan solution was taken out and placed in a polytetrafluoroethylene mold. Pour 0.1 M phosphate buffer solution containing 0.45 M NaHCO₃ at a ratio of 1:0.7 (v/v) to the chitosan solution into the chitosan solution, and stir quickly and evenly with an electric stirrer. Stand still for 30 min to wait for chitosan solid phase separation and precipitation as solid. The chitosan sponge after phase separation was taken out and soaked in PBS for 10 min to wash off excess salt. The chitosan sponge after phase separation is taken out and placed in a mold, and placed in a −20 °C low-temperature refrigerator for 30 min to freeze to 0 °C, and the temperature was monitored and verified using a temperature probe.

The frozen and phase-separated chitosan sponge was taken out, and freeze-dried for 8 h to obtain a dry phase-separated chitosan sponge (spCS). A dichloromethane (DCM) solution of dodecanal with a chitosan sponge amino molar mass of 0.025 eq. was prepared, and the superporous hemostatic sponges were soaked in the solution to extrude bubbles, then taken out, and left to react for 5 min. Add the reacted dodecyl chitosan to a DCM saturated solution of sodium triacetoxyborohydride with a chitosan amino molar mass of 4 eq to reduce the unstable carbon-oxygen double bond to a stable carbon-oxygen bond, shaker at 100 rpm to restore overnight. Chitosan was soaked in 95% ethanol for 6 h to remove unreacted sodium triacetyl borohydride, DA, and DCM. Soak in deionized water for 1 h to remove excess ethanol. Freeze at −20 °C for 30 min and freeze-dry to obtain an alkylated-modified phase-separated chitosan sponge (A-spCS). The phase separation chitosan sponge (pCS) subjected the chitosan solution to initial phase separation and −80 °C treatment before proceeding with freeze-drying. To use a chitosan sponge with the same solid content without phase separation, the control group used 1.5 wt% chitosan solution, which was freeze-dried to obtain a chitosan sponge without phase separation and secondary network compaction (CS).

### Micro-CT analysis
Quantum GX2 Analyze (version 14.0) was used to analyzed micro-CT data and network density.

### XPS analysis
X-ray photoelectron spectroscopy (Escalab Xi+, US) was used to detect the surface chemical structures of chitosan sponges before and after alkylation grafting. N1s peaks were processed with AVANTAGE software (version: 5.52).

### Mechanical test
The prepared lyophilized CS, pCS, spCS, and A-spCS are made into a cylindrical sample with a diameter of about 8 mm. The universal testing machine (MTS, US) is used to measure the axial force. The total test distance is 85% of the height of the sample, and the stepping speed is 100 μm/s.

### Characterization of microstructure/ porosity of spCS and A-spCS
Macro and microstructures were characterized by scanning electron microscopy (Merlin, Germany). The average pore diameter was measured using ImageJ software (version: 1.53k). Samples containing blood cells were washed with PBS, fixed with 4% paraformaldehyde solution, and soaked in a series of gradient alcohol (40%, 60%, 80%, 100%) solutions for 10 min to remove water, and air-dried at 37 °C for 12 h. Cut with a scalpel after drying, spray platinum on the cut surface, and observe with a SEM.

The porosity of the sponges was measured by the n-hexane method. Immerse them in the hexane solution for 24 h at room temperature. The mass of the chitosan sponge after freeze-drying was weighed and recorded as $W_d$ (g). Measure their wet weight after removing excess hexane from the surface of the samples using damp filter paper, regarded as $W_w$ (g). Percent porosity is calculated by the following equation:

$$\text{Porosity} (\%) = \frac{W_w - W_d}{\rho V} \times 100\% \tag{1}$$

$\rho$ is the density of n-hexane, and $V$ is the volume of the chitosan scaffold.

### Characterization of water/blood absorption properties of superporous spCS and A-spCS hemostatic sponges
To quantitatively evaluate the absorption capacity, the mass of the chitosan sponge after freeze-drying was weighed and recorded as $W_d$ (g). The compressed CS, pCS, spCS, A-spCS, and epCS were then immersed in the water and blood of rats. The chitosan was taken out and weighed to obtain the mass, and the water absorption/blood volume of the chitosan sponge changed with time and was drawn according to the mass change. Then, weigh the mass of the sponge after absorbing water/blood for 1 min, and record it as $W_w$ (g). Calculate the water/blood absorption capacity according to the following equation:

$$\text{Ability of absorption} (g/g) = \frac{W_w - W_d}{W_d} \times 100\% \tag{2}$$

### The pro-coagulant ability of spCS and A-spCS hemostatic sponges
The pro-coagulant ability of the chitosan sponge was evaluated by measuring the coagulation index (BCI). 100 mg CS, pCS, spCS, A-spCS,

medical gauze, gelatin sponge, and CELOX-E hemostatic gauze are compressed to squeeze out water and placed in the EP tube, and CELOX hemostatic powder is placed directly in the EP tube. After incubation at 37 °C for 10 min, 100 μL of rat heparinized whole blood at 37 °C was dropped onto its upper surface (300 μL blood was used for high-volume procoagulant ability test). After incubating at 37 °C for 5 min, add 10 mL of deionized water to each EP tube, soak for 1 min, and use a UV spectrophotometer to measure the optical density (OD) of the supernatant at 540 nm, recorded as $OD_h$. A solution of mixed deionized water and rat heparin sodium whole blood (10 mL/100 μL) was used as a negative control, recorded as $OD_n$, and BCI was calculated according to the following equation:

$$BCI\,(\%) = \frac{OD_h}{OD_n} \times 100\% \qquad (3)$$

## Blood cell adhesion ability of spCS and A-spCS hemostatic sponges

The blood adhesion ability of the chitosan sponge was determined by measuring the 1-hour adhesion of red blood cells on the material. 50 mg of CS, pCS, spCS, A-spCS, gauze, gelatin sponge, CELOX-E gauze, squeezed to squeeze out the water was placed in the EP tube, and CELOX hemostatic powder was placed directly in the EP tube. Drop 100 μL of RBC suspension (300 μL RBC suspension was used for high-volume blood cell adhesion ability test) onto their top surface. Incubate at 37 °C for 1 h. Rinse three times with phosphate buffered saline (PBS, pH = 7.4) to remove non-adhered RBCs, then transfer to 4 mL of deionized water to lyse adhered RBCs to release hemoglobin for 1 h. Take 100 μL of supernatant and place it in a 96-well microplate, and measure its OD value ($OD_h$) at 540 nm. The OD 540 nm value of the solution composed of 100 μL RBCs suspension and 4 mL deionized water was used as the negative control ($OD_n$), and the RBC adhesion ability was calculated by the following formula:

$$RBCs\ adhesion\,(\%) = \frac{OD_h}{OD_n} \times 100\% \qquad (4)$$

Platelet adhesion was determined by a similar method. The material was compressed to squeeze out moisture and placed in a 24-well microplate. Then, 100 μL (300 μL RBC suspension was used for high-volume platelets adhesion ability test) of platelet-rich plasma (PRP) was dropped on its top surface, followed by incubation at 37 °C for 1 h. Next, they were washed with PBS to remove non-adhered platelets and soaked in 1% TritonX-100 solution to dissolve the platelets to release lactate dehydrogenase (LDH). After treatment with LDH kit (beyotime, China), the OD 490 nm value of the supernatant was measured and recorded as $OD_h$. 100 μL of PRP not exposed to hemostatic agent was used as a negative control, recorded as $OD_n$. The percentage of adherent platelets was calculated by the following equation:

$$Platelet\ adhesion\,(\%) = \frac{OD_h}{OD_n} \times 100\% \qquad (5)$$

## Antibacterial capability tests

*Escherichia coli* (*E. coli*), *Staphylococcus aureus* (*S. aureus*), and *Pseudomonas aeruginosa* (*P. aeruginosa*) are propagated and expanded in liquid lysate broth (LB). Tissue culture plates, gauze, gelfoam, CELOX, CELOX-E, CS, and pCS were used as controls. Before testing, spCS and A-spCS were compressed to squeeze out water and placed in a 24-well microplate. After being sterilized under ultraviolet light for 1 h, the bacterial suspension (10 μL, $10^8$ CFUs/mL) was dropped on the upper surface of each group of materials. Incubate at

37 °C for 2 h. Add 2 mL of sterile PBS buffer to each well to resuspend viable bacteria. Next, remove 20 μL of the resuspended bacterial suspension and dilute to 50 mL to obtain the final diluted bacterial suspension. Subsequently, spread 20 μL of the resuspension on the surface of the LB agar plate and incubate at 37 °C for 12 h. Count the CFU formed on each LB agar plate. Count the CFU formed on each LB agar plate and record it as $C_e$. Among them, the number of CFU in the group not in contact with the material is $C_n$. For groups with CFU below the limit of detection (LOD), the CFU count is taken as the LOD divided by the square root of 2. Bacterial log reduction was calculated by the following formula:

$$Bacterial\ log\ reduction = \log C_n - \log C_e \qquad (6)$$

## In vivo hemostasis test in rat model

The experimental operations in this study strictly complied in accordance with the appropriate ethical guidelines, and the use of animals in this study was approved by the South China University of Technology Animal Care and Use Committee (Permit Number 2022032). The rat (female, 250−300 g, 7−8 weeks) hepatic hemorrhage model was used to evaluate the hemostatic effects of CS, spCS, A-spCS, gauze, gelatin sponge, CELOX powder, and CELOX-E hemostatic gauze. Anesthetize the rat with 2% sodium pentobarbital solution (50 mg kg⁻¹). Expose the liver of the rat through an abdominal incision and place a pre-weighed filter paper under the liver. After resecting a piece of liver with a 6 mm circular cutting edge, different hemostatic materials were put into the wound to stop bleeding. Among them, the cylindrical (diameter of 8 mm) spCS and A-spCS were soaked in PBS, squeezed out water, and then put into the wound for hemostasis. Weigh and calculate the blood loss before and after the filter paper absorbs blood. Given that the hemostatic response to non-compressive wounds theoretically exhibits no significant gender disparities, gender-specific analyses were not pursued in our study.

## In situ liver regeneration

The in situ pro-regenerative capacities of CELOX, CS, pCS and A-spCS were evaluated using a representative rat liver defect model. Rats (female, 250−300 g, 7−8 weeks) were anesthetized with 2 wt% sodium pentobarbital. A-spCS was compressed and filled into the 6 mm penetrated liver wound. After hemostasis, the abdomen was sutured, painkillers were given for 3 days, and normal feeding was carried out for 4 weeks. After the rats were sacrificed, the livers were removed, fixed, and sectioned for H&E staining and immunofluorescence staining. Tissue ingrowth was assessed by H&E staining. Cell infiltration was assessed by DAPI staining. Immunofluorescence staining for von Willebrand factor (vWF) (Proteintech) and albumin (ALB) (Proteintech) at dilutions of 1:400 and 1:800 was performed to assess vascularization and hepatocyte infiltration. Immunofluorescent staining for hepatocyte nuclear factor (HNF-4α) (Affbiotech) was performed at a dilution of 1:250 to assess the expression of hepatic cytokines. CoraLite 488-conjugated Recombinant Goat Anti-Mouse IgG (Proteintech) and CoraLite 488-conjugated Recombinant Goat Anti-Rabbit IgG (Proteintech) were diluted 1:200. Observe and acquire images using a fluorescent inverted microscope (Thermo Fisher).

## In vivo hemostasis test in minipig model

The experimental operations in this study strictly complied in accordance with the appropriate ethical guidelines, and the use of animals in this study was approved by the South China University of Technology Animal Care and Use Committee (Permit Number 2022032). The hemostatic effect of CELOX and A-spCS was evaluated by a minipig liver hemorrhage model (female, 20−25 kg, 5 months). Bama minipigs were anesthetized using an intravenous bolus of propofol. After resecting a piece of liver with a 15 mm circular cutting edge, different hemostatic materials were put into the wound to stop bleeding. After bleeding, the compressed

cylindrical A-spCS (diameter of 18 mm) was squeezed out of water and filled into the wound cavity. Weigh and calculate the blood loss before and after the paper absorbs blood. Given that the hemostatic response to non-compressive wounds theoretically exhibits no significant gender disparities, gender-specific analyses were not pursued in our study.

### In vitro transport and migration experiments of cells in A-spCS

NIH 3T3 fibroblasts and LX-2 hepatic stellate cells were cultured in a culture dish for 24 h in high-glucose medium DMEM (5% fetal bovine serum albumin, 1% penicillin and streptomycin). Cells were trypsinized and resuspended in medium. NIH 3T3 fibroblast suspension ($1 \times 10^6$ per mL) was dropped on the upper surface of cylindrical A-spCS (8 mm in diameter and 8 mm in height), and A-spCS was placed in a culture dish containing medium. After 5 days of co-cultivation, A-spCS was cut vertically, and the cells were stained with Calcein dye (Beyotime, 1:1000). Use a fluorescence microscope (Thermo Fisher) to observe the number and morphology of the cells on the top, middle, and bottom of A-spCS.

### Statistics and reproducibility

Experiments were performed with triplicate samples ($n = 3$) and statistical analysis was performed using GraphPad Prism 9. All data are shown as mean ± standard deviation. Independent t-test and one-way ANOVA followed by Tukey's multiple comparison test were used to determine statistical significance between two or more groups, respectively. $p < 0.05$ was considered statistically significant.

### Reporting summary

Further information on research design is available in the Nature Portfolio Reporting Summary linked to this article.

## Data availability

All data supporting the findings of this study are available within the article and its supplementary files. Any additional requests for information can be directed to, and will be fulfilled by, the corresponding authors. Source data are provided with this paper.

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

## Acknowledgements

This work was financially supported by Shenzhen Key Laboratory of Bone Tissue Repair and Translational Research (NO. ZDSYS20230626091402006, F.W.). This work was also supported by the Fundamental Research Funds for the Central Universities (2023ZYGXZR096, P.Z.), the Guangzhou Basic and Applied Basic Research Scheme (SL2023A04J00861, P.Z.), The National Natural Science Foundation of China (Grant No. 82272534, F.W.). Parts of Figs. 4a, 5a, 6a, 7a, Fig. 8a, g, Fig. S30 a, Fig. S31 and Fig. S32 were revised by using pictures from Servier Medical Art, by Servier, licensed under a Creative Commons Attribution 4.0 Unported License (https://creativecommons.org/licenses/by/4.0/).

## Author contributions

L. Bian, P. Zhao, and F. Wei conceived the idea. T. Jiang and P. Zhao developed the preliminary phase separation preparation method, and further improved and found the relevant preparation method for secondary network compaction. T. Jiang performed mechanical properties, Micro-CT, XRD, XPS, liquid absorption capacity, procoagulant ability, antibacterial ability characterization, and biocompatibility. T. Jiang, S. Chen and J. Xu designed and performed in vivo experiments and analysis. H. Fu, Y. Zhang, and Q. Ling performed scanning electron microscopy characterization. Y. Xu, X. Chu, R. Wang, L. Hu, H. Li, and W. Huang helped in animal experiments. T. Jiang prepared the figures and wrote the manuscript with the input of all authors. L. Bian, P. Zhao, and F. Wei supervised the research.

## Competing interests

The authors declare no competing interests.

## Additional information

[1]School of Biomedical Sciences and Engineering, Guangzhou International Campus, South China University of Technology, Guangzhou 511442, China. [2]National Engineering Research Center for Tissue Restoration and Reconstruction, South China University of Technology, Guangzhou 510006, China. [3]Department of Orthopedic Surgery, The Seventh Affiliated Hospital of Sun Yat-sen University, Shenzhen 518107, China. [4]Department of Orthopedics, Union Hospital, Tongji Medical College, Huazhong University of Science and Technology, Wuhan 430022, China. [5]Department of Joint Surgery, First Affiliated Hospital of Sun Yat-sen University, Guangzhou 510080, China. [6]Guangdong Provincial Key Laboratory of Biomedical Engineering, South China University of Technology, Guangzhou 510006, China. [7]Key Laboratory of Biomedical Materials and Engineering of the Ministry of Education, South China University of Technology, Guangzhou 510006, China. [8]Shenzhen Key Laboratory of Bone Tissue Repair and Translational Research, Shenzhen 518107, China. [9]These authors contributed equally: Tianshen Jiang, Sirong Chen, Jingwen Xu. ✉e-mail: bianlm@scut.edu.cn; scutzpc1993@scut.edu.cn; weifuxin@mail.sysu.edu.cn

