## [Peer Review File · Nature Communications]

REVIEWER COMMENTS

Reviewer #1 (Remarks to the Author):

This manuscript reports the development of super porous chitosan sponge prepared by secondary network compaction to achieve enhanced permeability and mechanical properties for non-compressible hemostasis. The sponge materials were characterized in terms of their morphology, mechanical, physical (porosity and absorption) properties. In vitro antibacterial tests and in vivo hemostasis tests (rats and minipigs models) were also conducted, and the results were compared to those of commercial products. The methods were clearly described, and results discussed sufficiently.

However, I feel that the novelty of the reported work is not yet up to the journal of Nature Communication. The material used in this work, Chitosan, has been studied and developed extensively in the past few decades for biomedical applications including hemostatic wound dressings. There has also been work that explores the impact of freeze-drying temperatures on the structure, properties and performances of chitosan sponges/dressings (discussion on which could have been added to the literature review section of the manuscript). Since the strength of this manuscript is more on the characterization of the developed chitosan sponge including in vivo tests, I feel it would fit better in another journal in biomaterials.

Reviewer #2 (Remarks to the Author):

The authors developed superporous chitosan sponge (spCS) and alkylated superporous chitosan sponge (A-spCS) by using the temperature-assisted secondary network compaction strategy following the phase separation-induced primary compaction. This method has certain innovation, and the material presents different from the ordinary freeze-dried porous sponge material, showing improved mechanical properties and higher porosity. Based on the shape recovery and procoagulant properties of such materials, they are intended to be used for the rapid hemostatic first aid of non-compressible bleeding. However, from the perspective of hemostatic application, there are still some shortcomings in the research of this material, and the morphology structure and hemostatic performance seem to lack advantages. Besides, there are some deficiencies in the study of the sponges in the present manuscript. The specific comments are shown below.

1. Why is the structure and performance of the sponges best at 0°C? Only -80°C was used as a control below 0°C. What is the reason for this optimal temperature of 0 °C ? Is there a better choice between -80°C and 0°C?
2. When modifying the sponge by using hydrophobic alkane chains, how to control the amount of the hydrophobic alkane chains? In theory, this modification may increase the hydrophobic properties of the material and reduce the wettability. There is a lack of evidence to explore the effects of this hydrophobic chain and its specific contribution to hemostasis.
3. What is the shape fixation principle of the sponge? It doesn't seem to fix well.
4. The antibacterial effect of the material is best presented as a logarithmic reduction. The antibacterial effect seems to be very ordinary. Does this meet the usage requirements?
5. In the sponge preparation part, what is the meaning of freezing to 0°C at -20 °C? How long? Does this freeze-drying parameter also affect the properties of the material? The experimental method is not clear.
6. In fact, in compression testing, although under high compressive strain, it has a high compressive strength, the compressive strength is relatively low within the first 60% of the compressive strain range. What is the shape fixation ratio of the materials? In the application of expansion hemostasis, it is likely that the material has insufficient compressive strength under low deformation, making it difficult to expand to a large proportion, leading to the insufficient hemostasis effect and requiring a large number of materials.
7. Judging from the macroscopic photos, the macropores of the material are very large and the structural uniformity is not very good. Based on previous research results, this particularly large hole may not be conducive to promoting coagulation. This may also be the reason for its low compression strength at low compressive strains. Therefore, it seems that the advantages of sponges made by this method in hemostasis are not obvious.
8. Although the manuscript emphasizes its good mechanical strength, its characterization mainly involves compression testing and cyclic compression tests. In fact, hemostatic material is a one-time application, and the good stability of cyclic compression is of less significance. If its tensile strength is high, such as X-STAT material, it will be easier removed after use, with less residue. Because, chitosan has poor biodegradability and needs to be removed when used in large quantities.
9. The authors also emphasize the material's repair properties for liver tissue. Chitosan degrades poorly. For liver tissue with good self-repairing function, will it occupy space and hinder liver regeneration? As a tissue repair material, its in vivo degradation needs to be evaluated. In addition, why is the material perform better than gelatin sponge with better biocompatibility?
10. A large number of shape-memory hemostatic sponges have been reported, and their hemostatic effects have been verified in fatal arterial hemorrhage in large animals. The hemorrhage models have many options for effective hemostasis and are not challenges in first aid. The control hemostatic materials used do not rely on expansion to stop bleeding, so they cannot reflect the hemostatic advantages of the reported material more clearly. The authors need to compare the hemostasis advantages of this material with the reported shape-memory or expandable hemostatic materials.

11. The manuscript also does not mention how much squeezing force the material can provide during expansion and hemostasis. This squeezing force also seems to be critical for hemostatic effect that relies on expansion properties.

Reviewer #3 (Remarks to the Author):

The paper by Jiang et al. describes a new material for the treatment of non-compressible haemorrhages, which is an improvement of previously known material, chitosan. By using a specific pre-freezing temperature in the preparation of the material and coating its surface with hydrophobic dodecyl chains, the authors achieve improved pore size, better interconnection of the porous network, increased mechanical properties and improved coagulation ability of the material – which they name “superporous chitosan sponges” – spCS or “alkylated spCS” – A-spCS. The authors then explore the infiltration of spCS and A-spCS by red blood cells, platelets, coagulation and infection potential, and finally explore the material in the models of liver or liver/spleen injury in rats and mini-pigs, respectively. There is also a mention of scalability potential for the production of this material, which is essential if this material was to be used in the clinic.

The work constitutes a potential advance for the treatment of non-compressible haemorrhage. However, there are several issues which authors should address before this paper can be published in Nature Communications.

Major comments:

1. In the investigation of strain-stress on the different materials, the authors describe better qualities of the spCS and A-spCS. Could the authors either investigate experimentally or comment on how the range described in the experiments correlates to actual values in human treated non-compressible haemorrhage?
2. In paragraph about pro-coagulation, the authors describe that both spCS and A-spCS are superior to other materials tested. While I agree with A-spCS being superior in the 3 methods tested, spCS only performed better in 1/3 tests used by the authors in Fig. 4 – the authors should tone down their statements here or provide additional data about spCS.
3. In the test of anti-bacterial properties, could the authors comment on the E.coli and S.aureus infection prevalence after non-compressible haemorrhage? Are there any other bacterial infections known to be prevalent after this type of injury that could be relevant and tested here? Finally, could the authors comment on the 2h incubation with the bacteria in relation to real life situation, where the material might be exposed for longer to contamination?

4. The authors should use another type of cells instead of 3T3 embryonic fibroblasts for testing compatibility of cells with their material, which is more relevant to the cells encountered in this type of damage. A good choice here would be primary liver or splenic cells, or at least liver/spleen cell line.

5. The investigation of liver cell infiltration in Fig.8 is insufficient, and the following should be corrected:

- Could the authors better outline which part of the tissue is the materials tested and which is the liver? While I can guess, it would be better to specify this in the figure itself. As it is, I am not convinced that A-spCS is better than the other materials tested in this experiment.

- the pictures in Fig.7b are of insufficient quality to assess the infiltration of different cells into the material. Could the authors improve the resolution? How exactly was the quantification performed?

- HNF4a is not a cytokine, but a nuclear transcription factor – please correct lines 415-416 and the whole sentence for accuracy.

- Finally, Fig.8g should be performed with liver-relevant cells, and crucially - also comparing migration to other materials tested in the paper, and not shown in isolation as it is now. If authors want to show relevance for the migratory ability of liver fibroblast cells, the use of primary cells or at least hepatic stellate cell line should be employed here.

Minor comments:

- For the live/dead staining, authors only show Calcein live stain. Was any dead stain used? How were the cells counted for the assessment of viability? If no dead cell stain was used, the authors should correct this in the text.

- Could the authors please include a title in each of their supplementary videos? It is very hard to follow the different materials and techniques as it is, without any title embedded within the videos.

- In figure 3, panels k, l and m are very hard to decipher; perhaps use of different colours could be implemented by the authors here to improve clarity or simply a bigger panel with higher resolution?

Response to the Comments from the Reviewers:

We appreciate the reviewers for positive comments and the feedbacks provided to strengthen the manuscript. We provide a point-by-point response to each of the reviewers. The corresponding changes to the revised Manuscript and Supporting Information have been highlighted in yellow.

Reviewer #1:

This manuscript reports the development of super porous chitosan sponge prepared by secondary network compaction to achieve enhanced permeability and mechanical properties for non-compressible hemostasis. The sponge materials were characterized in terms of their morphology, mechanical, physical (porosity and absorption) properties. In vitro antibacterial tests and in vivo hemostasis tests (rats and minipigs models) were also conducted, and the results were compared to those of commercial products. The methods were clearly described, and results discussed sufficiently.

However, I feel that the novelty of the reported work is not yet up to the journal of Nature Communication. The material used in this work, Chitosan, has been studied and developed extensively in the past few decades for biomedical applications including hemostatic wound dressings. There has also been work that explores the impact of freeze-drying temperatures on the structure, properties and performances of chitosan sponges/dressings (discussion on which could have been added to the literature review section of the manuscript). Since the strength of this manuscript is more on the characterization of the developed chitosan sponge including in vivo tests, I feel it would fit better in another journal in biomaterials.

Response:

We appreciate the reviewer for the positive recognition on our comprehensive characterizations of this super porous chitosan sponge prepared by secondary network compaction. We have further conducted more detailed in vitro and in vivo studies to demonstrate the superiority of our chitosan sponge with enhanced permeability and mechanical properties as requested by other reviewers (details are provided in the Response to the comments of other reviewers). More importantly, we'd also like to highlight the significance of our work besides the comprehensive characterizations of the biomaterials in a

more explicit manner as below to address Reviewer #1's concern on the novelty of our work.

*Firstly, in pursuit of achieving the ultimate clinical translation, biomaterials researchers usually focus on a limited set of key building block biomaterials with promising prospects of FDA approval instead of keeping on developing new biomaterials, and chitosan is one of these building block biomaterials. Therefore, developing new fabrication strategies to address the inherent limitations of these biomaterials, such as weak mechanical strength and poor bio-infiltration, has been the key objective of biomaterial research in the recent years. For instance, Professor Ximin He has developed simple yet highly effective strategies to fabricate robust **PVA** hydrogels through the simple freeze casting and salting-out and to prepare high-permeability robust **polyacrylamide** hydrogels through one-step phase separation¹⁻⁴. Professor Michael D. Dickey also demonstrated the fabrication of tough and stretchable ionogels by in situ phase separation of **acrylamide** and **acrylic acid**⁵. Moreover, Professor Jianping Gong, Zhigang Suo and Feng Jiang prepared strong toughness hydrogels by controlling the pre-stretching of **alginate**, **gelatin**, and **poly (acrylic acid)** to induce network alignment, forming a dense entanglement of 2D hierarchical structures with polymer chains in a layered fiber structure⁶⁻⁹. These studies on network restructuring and strengthening of conventional biomaterials have been published in leading multi-disciplinary journals including *Nature*, *Nature Materials*, and *Advanced Materials* and attracted interest and attention from researchers in diverse research areas. Such endeavors represent significant advancements in this field, offering innovative yet simple approaches to the design and fabrication of advanced polymeric devices by simply manipulating the network structures of these conventional biomaterials.*

Secondly, despite the extensive literatures on chitosan biomaterials to date, fabricating biomedical chitosan sponge with combined superporosity and mechanical resilience remains a challenge. Our strategy focuses on addressing the trade-offs between pore connectivity and mechanical performance inherent in the traditional preparation methods of chitosan sponge (Fig. S1). This strategy offers a straightforward and generalizable method to facilitate the the scale-up production and eventual clinical translation by providing a scalable and biocompatible network restructuring strategy without using any complicated chemical modifications.

Despite several reports on temperature effects on chitosan sponges based on freeze-drying in the past, the temperature-assisted secondary network compaction (TA-2nd NC) strategy

diverges markedly from the prior methods, both in terms of procedure, principles and outcomes. In prior reports, chitosan solutions were directly freeze-dried under conditions of -20°C, -80°C, and -196°C to form chitosan sponges. The sponges prepared using ice crystals as pore templates, exhibited pore sizes ranging from 10 to 250 μm, with porosities of 23% to 80%, limited water absorption ability (absorb 5.5 - 9.5 g/g water in 24 hours) and experienced a decrease in maximum stress to approximately 87% after 10 cycles¹⁰⁻¹³. In contrast, we first induced phase separation through pH modulation to induce the primary network compaction, followed by the secondary network compaction during freeze-drying at specific pre-freezing temperatures (-80°C, 0°C, 20°C). At the optimal pre-freezing temperature, ice crystals primarily acted as restrictors of network motion to prevent excessive network restructuring and damage to the chitosan sponge structure. Compared to the traditional chitosan sponge (CS) fabricated by the direct freeze drying, spCS prepared by our TA-2nd NC strategy possesses enhanced sponge framework compaction density (83% higher than CS), high sponge porosity (460% higher than CS), rapid water-triggered shape recovery (0.85 s), exceptional water absorption capability (122% higher than CS), and outstanding mechanical performance (retains 95% of the maximum stress after 100 cyclic compression at 85% strain).

These data demonstrate the efficacy of our TA-2nd NC strategy to address the challenge of combining high porosity and mechanical performance, two seemingly mutually exclusive attributes, faced by the traditional preparation methods of chitosan sponge. Without this concept, it would have not been possible to achieve the outstanding hemostatic effect in our chitosan sponges. We have added this detailed discussion in the revised manuscript to more explicitly elaborate on the novelty and importance of this new approach.

Fig. S1. Preparation strategies differences and properties differences between traditional strategy and our temperature-assisted secondary network compaction (TA-2ndNC) strategy. Traditional preparation strategies of chitosan sponge show significant trade-offs between pore connectivity and mechanical performance. In contrast, our TA-2ndNC strategy achieves outstanding hemostatic effect in our superporous chitosan sponges, which are a combination of mutually incompatible properties, i.e. high network density, high network porosity, rapid water-triggered shape recovery, exceptional water absorption capability, and outstanding mechanical performance.

Reviewer #2:

The authors developed superporous chitosan sponge (spCS) and alkylated superporous chitosan sponge (A-spCS) by using the temperature-assisted secondary network compaction

strategy following the phase separation-induced primary compaction. This method has certain innovation, and the material presents different from the ordinary freeze-dried porous sponge material, showing improved mechanical properties and higher porosity. Based on the shape recovery and procoagulant properties of such materials, they are intended to be used for the rapid hemostatic first aid of non-compressible bleeding. However, from the perspective of hemostatic application, there are still some shortcomings in the research of this material, and the morphology structure and hemostatic performance seem to lack advantages. Besides, there are some deficiencies in the study of the sponges in the present manuscript. The specific comments are shown below.

1. Why is the structure and performance of the sponges best at 0°C? Only -80°C was used as a control below 0°C. What is the reason for this optimal temperature of 0 °C? Is there a better choice between -80°C and 0°C?

Response:

We appreciate the reviewer's constructive comments, which have greatly enhanced the quality of our manuscript. To address the reviewer's comment, we have prepared chitosan sponges at the pre-freeze drying temperatures (T_{pfd}) of -40, -20, -5, and 5 °C and characterized their material properties. Scanning electron microscopy (SEM) and porosity tests revealed that with higher T_{pfd} , the pore size and porosity gradually increased (Fig. S3 and Fig. S4). This can be attributed to the alleviation of constraints on the mobility of chitosan network at higher freezing temperatures, thereby enhancing the extent of secondary network restructuring induced by TA-2nd NC strategy.

Fig. S3. Scanning electron microscope (SEM) images of the microstructure of sponges prepared at pre-freeze drying temperatures of 5, 0, -5, -20, and -40 °C. The spCS prepared at

T_{pfd} of 0°C exhibits a highly interconnected superporous structure.

Fig. S4. Porosity and pore size of sponges prepared at pre-freeze drying temperatures of 5, -5, 0, -20, and -40 °C. As T_{pfd} increases, both porosity and pore size gradually increase. The spCS prepared with T_{pfd} set at 0°C exhibits higher porosity and larger pore sizes compared to the -20°C and -40°C groups. a) Porosity of sponges prepared at T_{pfd} of -40, -20, -5, 0, and 5°C. b) Pore size of sponges prepared at T_{pfd} of -40, -20, -5, and 5°C (n=25).

An ideal hemostatic sponge with high liquid permeability can achieve rapid shape recovery upon liquid contact, thereby establishing a physical barrier on wound. Therefore, we analyzed the dynamic curves of water absorption over time for sponges prepared at different pre-freeze drying temperatures. As T_{pfd} increased, both the capacity and rate of liquid absorption significantly increased (Fig. S8). However, these metrics of sponges prepared at other T_{pfd} remained lower than those of superporous sponge (spCS) prepared at 0°C T_{pfd} , which has the capacity to absorb water up to 23.2 times its own weight and can absorb water at a rate of 16.3 times its own weight per second.

Fig. S8. Water absorption ability of sponges prepared at T_{pfd} of -40, -20, -5, 0 and 5 °C. The spCS prepared with T_{pfd} set at 0°C exhibits the highest liquid absorption rate and capacity. Materials from different groups show a gradual decrease in liquid absorption capacity as T_{pfd} away from 0°C. a) Water absorption capacity of sponges prepared at T_{pfd} of -40, -20, -5, 0 and

5°C in 15 seconds. b) Water absorption capacity of sponges prepared at T_{pfd} of -40, -20, -5, 0 and 5°C. c) Water absorption rate of sponges prepared at T_{pfd} of -40, -20, -5, 0 and 5°C.

We also investigated the mechanical properties of sponges prepared under different pre-freeze drying temperatures. Cyclical compression testing indicated that the maximum stress of 100 cycles decreased to below 80% for the -40°C, -20°C, -5°C, and 5°C groups (Fig. S14). Among these, the -5°C group exhibited the best stress retention and minimal hysteresis loop, suggesting that sponges prepared at pre-freeze-drying temperatures closer to 0°C possess superior mechanical properties. In summary, sponges subjected to secondary network restructuring at pre-freeze drying temperatures approaching 0°C demonstrate higher porosity, enhanced liquid absorption capacity, and superior mechanical performance.

Fig. S14. Mechanical performance test of sponges prepared at T_{pfd} of -40, -20, -5, and 5 °C. The spCS prepared with T_{pfd} set at 0°C exhibits the highest stress retention rate and the lowest hysteresis loop ratio. Materials from different groups demonstrate a gradual decrease in mechanical performance as T_{pfd} away from 0°C. a-d) Stress-strain cyclic curves sponges prepared at T_{pfd} of -40, -20, -5, and 5°C. e) Maximum stress retention rates of sponges prepared at T_{pfd} of 40, -20, -5, 0, and 5°C after 100 times stress-strain cycles. f) Hysteresis loops of stress-strain curves sponges prepared at T_{pfd} of 40, -20, -5, 0, and 5°C.

2. When modifying the sponge by using hydrophobic alkane chains, how to control the amount of the hydrophobic alkane chains? In theory, this modification may increase the hydrophobic properties of the material and reduce the wettability. There is a lack of evidence to explore the effects of this hydrophobic chain and its specific contribution to hemostasis.

Response:

We appreciate the reviewer's constructive comments. The modification degree of hydrophobic alkyl chains was controlled by the feed ratio of dodecyl aldehyde to chitosan, and then monitored by XPS semi-quantitative analysis.

Theoretically, highly hydrophobic alkyl chains may reduce liquid infiltration. However, studies have reported that certain hydrophobic modification of gauze, rather than high hydrophobic modification ratio, can achieve a controlled balance between hydrophilicity and hydrophobicity and significantly enhance gauze's hemostatic ability¹⁴. Furthermore, it seems difficult for hydrophobic chains to cover the hydrophilic sites of the polysaccharide ring to reduce liquid absorption capacity in this work. The liquid absorption curves (Fig. 3d, g) confirm that chitosan modified with alkyl chains did not weaken its liquid absorption capacity.

Therefore, we summarized the effects of hydrophobic alkyl chains on hemostasis through three aspects. Firstly, the adhesion of hydrophobic alkyl chains to red blood cells can aggregate a large number of red blood cells^{15,16}, as confirmed by the red blood cell attachment experiment (Fig. 4b). Red blood cells are an important component of thrombi, and the accumulation of a large and thick layer of red blood cells facilitates the formation of large clots, thereby promoting hemostasis. Secondly, the adhesion of hydrophobic alkyl chains to platelets (Fig. 4e) can activate the cascade hemostasis in vivo¹⁷, thereby accelerating the formation of blood clots. Finally, in materials with a small amount of alkyl chain modification, the regulation of hydrophilicity and hydrophobicity can be achieved¹⁴. Blood can be absorbed smoothly, and the entry of blood into the sponge can be prevented from further seeping outwards, which mainly attributed to the restriction of the blood cells by alkyl chains, inhibiting further unfavorable diffusion leading to increased bleeding. Additionally, in terms of outcomes of hemostasis experiments conducted on rat liver penetrating injuries, we consider that hydrophobic modification facilitates hemostasis in vivo (Fig. 6).

*Fig. 6. In vivo hemostatic effect of A-spCS in SD rats. a) Schematic diagram of the rat liver perforation model and the hemostasis process. b) Photographs of hemostatic effects of blank, gauze, GS, CELOX, CELOX-E, CS, pCS, spCS, and A-spCS. The yellow dotted line is the boundary of the liver. c, d) Total blood loss and hemostatic time of blank, gauze, GS, CELOX, CELOX-E, CS, pCS, spCS, and A-spCS groups. Data are presented as the mean \pm SD. $P > 0.05$ (ns), $**P < 0.01$, $***P < 0.001$, $****P < 0.0001$ (one-way ANOVA or two-tailed Student's t-test to compare multiple groups or two groups).*

3. What is the shape fixation principle of the sponge? It doesn't seem to fix well.

Response:

We appreciate the reviewer's constructive comments and have thoroughly discussed the points raised by the reviewer. Due to spCS's excellent elasticity and fatigue resistance properties, it exhibits some resilience after complete compression, which can be attributed to two reasons,

separately. First, spCS can not entirely expel water after compression, which provide some support to the network. Second, based on previous literature¹⁸, we propose that network of spCS includes large fiber skeleton composed by high density area and small fiber network composed by low density area as shown in Fig. 2b. The large fiber skeleton with higher network density, formed through secondary network compression, primarily provides load-bearing function and interconnects various pore structures, imparting elasticity to the material, hence allowing it to rebound to some extent after compression. The small fiber network with lower chitosan network density, which can provide crosslinking for the large fiber skeleton, conversely, exhibits weaker load-bearing capacity and cannot rebound after compression. The combination of these two kinds of fiber structures enables the material to compress but exhibit slight rebound.

Nevertheless, in practical applications, we can mitigate the material's rebound tendency clamping it with syringes or forceps after compression.

Fig. 2. a) Micro-CT images of the microstructure of chitosan (CS), phase-separated chitosan (pCS), spCS, A-spCS, and epCS. b) Density heatmaps of the chitosan networks of CS, pCS, spCS, A-spCS, and epCS.

4. The antibacterial effect of the material is best presented as a logarithmic reduction. The antibacterial effect seems to be very ordinary. Does this meet the usage requirements?

Response:

We are grateful for the reviewer's constructive comments. We have modified the antibacterial effect to be presented as logarithmic reductions. Count the CFU formed on each LB agar plate and record it as C_e . Among them, the number of CFU in the group not in contact with

the material is C_n . For groups with CFU below the limit of detection (LOD), the CFU count is taken as the LOD divided by the square root of $2^{19,20}$. Bacterial log reduction was calculated by the following formula:

$$\text{Bacterial log reduction} = \log C_n - \log C_e$$

Typically, a reduction of bacterial percentage by 90% or more in 6 hours is considered indicative of strong antibacterial activity, which is equivalent to a lethality rate of 99.9% in 24 hours²¹. Two-hour antibacterial experiments demonstrate that A-spCS achieves a 96.1% reduction in *Escherichia coli* (*E. coli*) and 99.0% reduction in *Staphylococcus aureus* (*S. aureus*). We also investigated the antibacterial capability of A-spCS against *Pseudomonas aeruginosa* (*P. aeruginosa*), which is a common pathogenic bacterium known to cause various infections²², achieving an over 99% reduction in bacterial percentage (Fig. S23). Additionally, our four-hour and six-hour antibacterial experiments reveal that A-spCS achieves over 99% antibacterial efficacy against three bacterial strains (Fig. S24). Furthermore, our superporous chitosan sponge, as a highly porous carrier rich in amino groups, can effectively enhance its long-term antibacterial capability by simply attracting or chemically grafting various antibacterial advantageous groups or molecules. For instance, we loaded antibiotics into the porous network of A-spCS. By loading penicillin G into the chitosan network, we effectively achieved sustained antibacterial activity for 12 hours, 24 hours, and 48 hours (Fig. S25).

Fig. S23. In vitro anti-*P. aeruginosa* properties of various hemostatic materials. a) Photographs of colonies of *P. aeruginosa* grown on LB agar plates after contact with blank, gauze, GS, CELOX, CELOX-E, CS, pCS, spCS, and A-spCS. b) Corresponding statistical results of colony counts of *P. aeruginosa*.

Fig. S24. The evaluation of the antimicrobial efficacy of A-spCS after co-incubation for 4 and 6 hours. a) Photographs of colonies of *E. coli*, *S. aureus*, and *P. aeruginosa* grown on LB agar plates after contact with blank and A-spCS for 4 h b) and 6 h. c) Corresponding statistical results of colony counts for 4 h d) and 6 h.

Fig. S25. The evaluation of the antimicrobial efficacy of penicillin G-loaded chitosan sponge for 12, 24, and 48 hours. a) Absorbance of LB after incubation of *E. coli*, *S. aureus*, and *P. aeruginosa* with penicillin-loaded chitosan sponge for 12, 24, and 48 hours. b) Penicillin G sustained release curve.

5. In the sponge preparation part, what is the meaning of freezing to 0°C at -20 °C? How long? Does this freeze-drying parameter also affect the properties of the material? The experimental method is not clear.

Response:

We are grateful for the reviewer's meticulous review. We have revised the sponge preparation section. Indeed, rapid cooling can affect the properties of the material. For instance, rapid cooling methods such as rapid freezing in liquid nitrogen can lead to uneven ice formation and ice crystal expansion, which can disrupt the network structure. Therefore, we opted for a gentler cooling process. Specifically, after phase separation, the chitosan hydrogel was placed in a -20°C freezer for 30 minutes to freeze to 0°C, and the temperature was monitored and verified using a temperature probe.

Revised version:

The chitosan sponge after phase separation is taken out and placed in a mold, and placed in a -20°C low-temperature refrigerator for 30 minutes to freeze to 0°C, and the temperature was monitored and verified using a temperature probe.

6. In fact, in compression testing, although under high compressive strain, it has a high compressive strength, the compressive strength is relatively low within the first 60% of the compressive strain range. What is the shape fixation ratio of the materials? In the application of expansion hemostasis, it is likely that the material has insufficient compressive strength under low deformation, making it difficult to expand to a large proportion, leading to the insufficient hemostasis effect and requiring a large number of materials.

Response:

We appreciate the reviewer's constructive comments. The reviewer's feedback has prompted us to delve deeper into certain aspects of our research, resulting in a more comprehensive analysis.

We supplemented the testing regarding the shape fixation ratio. The formula for calculating the shape fixation ratio is as follows²³:

$$\text{Shape fixation ratio} = \frac{\varepsilon_{\text{fixed}}}{\varepsilon_{\text{max}}} \times 100\%$$

ε_{max} represents the maximum deformation and $\varepsilon_{\text{fixed}}$ represents the strain at which the material fixes its shape from the initial state. A shape fixation ratio closer to 100% indicates a higher degree of material fixation.

spCS cannot achieve completely shape fixation, which may be due to its exceptional anti-fatigue ability and elasticity. Thus, we tested the stress values under shape fixation state and at 85% strain, demonstrating that spCS can provide stress exceeding 100 kPa. Certainly, while our material exhibits lower strength at a compression rate of 60%, under our actual shape fixation state, the compression ratio exceeds 60%. In this condition, our material can provide a stress of 32.8 kPa (Fig. S20). Moreover, in practical applications, we can immobilize the material using syringes or forceps to reduce rebound after compression, which provide a higher compression ratio for the sponges.

Furthermore, in the rat liver puncture model hemorrhage experiment, the radius of the materials used was 4/3 times the radius of the wound, meaning the volume compression ratio of the material within the wound was $1 - 0.75 \times 0.75 = 43.75\%$. At this compression ratio, spCS can provide a stress of 6.81 kPa, while commercially available gelatin sponge can only provide a stress of 0.22 kPa. Moreover, spCS still achieved effective hemostasis and demonstrated a hemostatic efficiency higher than that of the control group (Fig. 6). This indicates that even when the compression ratio is lower than 60% or shape fixation ratio, the material still achieves efficient hemostasis in non-compressible wounds.

Fig. S13. Shape fixation ratio of CS, pCS, spCS, and epCS. a) A schematic diagram illustrating the compressed state and shape fixation state. b) Statistical graph depicting the shape fixation ratio.

Fig. S20. Compressive stress under different strain of gelatin sponge, CS, pCS, spCS, epCS. a) Compressive stress under 85% strain of gelatin sponge, CS, pCS, spCS, and epCS. b) Compressive stress under shape fixation state of gelatin sponge, CS, pCS, spCS, and epCS.

7. Judging from the macroscopic photos, the macropores of the material are very large and the structural uniformity is not very good. Based on previous research results, this particularly large hole may not be conducive to promoting coagulation. This may also be the reason for its low compression strength at low compressive strains. Therefore, it seems that the advantages of sponges made by this method in hemostasis are not obvious.

Response:

We appreciate the reviewer's constructive comments.

We consider that the hemostatic function of the hemostatic sponge primarily involves the pressure exerted on the wound through rapid liquid absorption-induced shape expansion and the confinement of blood cells within the sponge to form a clot²⁴⁻²⁶. Firstly, regarding the ability to rapidly absorb liquid, we consider that the sponge's heterogeneous pore structure with large pores (Fig. 2c) of approximately 1052.0 μm (Fig. 2f) and small pores of approximately 14.8 μm (Fig. S5) facilitates rapid blood absorption. According to the Washburn theory²⁷, the significantly pronounced capillary action in smaller pores provides the driving force for liquid movement and serves as connection points for liquid transport in larger channels. The high connectivity of large pores offers efficient pathways for liquid transport²⁸. Previous studies have shown that the transport efficiency of this mixed pore structure, combining large and small pores, is even higher than the sum of the liquid transport speeds of materials with single large or single small pores^{29,30}. The liquid absorption experiments also validate the outstanding liquid absorption capacity and shape recovery rate of spCS.

Fig. S5. Pore diameter of secondary channel spCS and A-spCS (n=25).

Fig. 2. c) Scanning electron microscope (SEM) images of the microstructure of CS, pCS, spCS, A-spCS, and epCS.

Secondly, while it is true that larger pore sizes often involve trade-offs with mechanical performance, TA-2nd network compression strategy endeavors to rectify this concern. Larger pore size does not necessarily correlate directly with mechanical performance. According to the mechanical property experimental results, despite spCS (1052.0 μm) having larger pore sizes than CS (17.8 μm) and smaller pore sizes than epCS (1648.0 μm), the pressure it provides (123.7 kPa) is significantly stronger than that of CS (27.8 kPa) and epCS (34.36 kPa) (Fig. S20). This can be attributed to the increased chitosan network density (Fig. 2b) resulting from phase separation and secondary network compression at the optimal temperature, thereby enhancing the mechanical properties of the chitosan network.

Fig. S20. Compressive stress under different strain of gelatin sponge, CS, pCS, spCS, epCS. a) Compressive stress under 85% strain of gelatin sponge, CS, pCS, spCS, and epCS. b) Compressive stress under shape fixation state of gelatin sponge, CS, pCS, spCS, and epCS.

Based on the above, we consider that the heterogeneous structure of our material, comprising uneven large pores mixed with small pores, facilitates liquid infiltration and provides sufficient pressure on the wound. Our in vivo hemostasis experiments in rats have also confirmed that even at a compression ratio of 43.75%, spCS achieves efficient hemostasis (Fig. 6).

8. Although the manuscript emphasizes its good mechanical strength, its characterization mainly involves compression testing and cyclic compression tests. In fact, hemostatic material is a one-time application, and the good stability of cyclic compression is of less significance. If its tensile strength is high, such as X-STAT material, it will be easier removed after use, with less residue. Because, chitosan has poor biodegradability and needs to be removed when used in large quantities.

Response:

We appreciate the reviewer's constructive comments and have made improvements to the study accordingly.

Our compression and cyclic compression testing aimed to validate the material's mechanical performance primarily for two purposes: the ability of the material to recover its shape and provide pressure after being subjected to compression, and the stable use of the material in different scenarios. Firstly, despite epCS possessing a highly interconnected large porous structure, its excessive network restructuring led to a decrease in toughness, resulting in the inability to recover its shape after compression, and a significant decrease in both overall

liquid absorption capacity and provided stress. Conversely, spCS and A-spCS were able to rapidly expand and provide sufficient pressure after compression, based on their excellent fatigue resistance properties. Secondly, while materials may undergo proper storage and transportation in conventional civilian settings, in emergency or battlefield situations, transportation and usage of materials may occur in more adverse and demanding environments. Therefore, the ability to maintain good performance under external influences remains an indicator for assessing material usability.

Furthermore, we conducted tensile testing on our material and demonstrated that spCS subjected to secondary network compaction exhibited higher tensile strength (76 kPa) (Fig. S15). Nonetheless, we deem material removal to be less favorable. Despite the numerous advantages of XStat, inserting approximately 92 micro-sponges into an open wound may lead to a 22-fold increase removal time compared to traditional gauze, as each sponge bed needs to be extracted individually³¹. This may cause discomfort to patients during prolonged device removal and increase surgical time and costs. Therefore, we consider that porous materials serving as tissue regeneration scaffolds may present a more feasible alternative. Due to its highly interconnected superporous structure with large pore size and high porosity, A-spCS facilitates cell migration, which is important to vascular development and tissue repair (Fig. 8). Additionally, in our supplementary subcutaneous degradation experiment (Fig. S30), we observed partial degradation of chitosan sponge after 4 weeks. The chitosan network degraded, while the large fibrous scaffold continued to provide overall structural support and create a protective physical microenvironment for cells during the liver's repair period, which ranges from several weeks to months.

Fig. S15. Tensile test of CS, pCS, spCS, and epCS.

Fig. S30. A-spCS and GS in vivo degradation experiment. a) Schematic diagram of in vivo explanation. b) Representative photos and HE staining pictures of degradation of A-spCS and GS.

9. The authors also emphasize the material's repair properties for liver tissue. Chitosan degrades poorly. For liver tissue with good self-repairing function, will it occupy space and hinder liver regeneration? As a tissue repair material, its in vivo degradation needs to be evaluated. In addition, why is the material perform better than gelatin sponge with better biocompatibility?

Response:

We appreciate the reviewer's constructive comments.

Research indicates that the complete recovery time for the liver in healthy individuals post-surgery takes several weeks to months³². For patients undergoing surgeries related to conditions such as hepatitis or cirrhosis, the recovery time for the liver is further prolonged. Therefore, tissue scaffolds used for tissue regeneration typically require gradual degradation over a longer time frame. These scaffolds are designed to provide temporary support and structural integrity post-implantation, while also creating a conducive environment for surrounding tissues to grow and regenerate. Studies have indicated that decellularized extracellular matrix (dECM) scaffolds may degrade relatively quickly in vivo due to their mechanical brittleness, resulting in a premature loss of the physical and biochemical microenvironment for encapsulated cells³³.

In our supplementary subcutaneous degradation experiments (Fig. S30), we observed collapse of the chitosan network structure and degradation of most of the chitosan sponge, while the large fiber skeleton continued to provide certainly structural support. Moreover, due

to the TA-2nd NC strategy, spCS is endowed with a highly interconnected superporous structure and a high porosity, allowing for the migration and infiltration of internal organizations throughout the pore structure of spCS within two weeks. In contrast, gelatin sponges were almost completely degraded within 2 weeks without obviously cellular migration into the sponge, which may not be sufficient to support complete repair of penetrating liver injuries. Therefore, we consider that the relatively slower degradation rate of chitosan sponges may not be a disadvantage.

Furthermore, due to the excellent biocompatibility of chitosan, its presence in the body does not induce significant inflammatory reactions and can adequately induce cell migration and growth. We supplemented the cell compatibility experiment with gelatin sponges and demonstrated that our chitosan material is not significantly different from gelatin sponges in terms of cell compatibility (Fig. S26). This may be attributed to the complete absence of chemical cross-linking agents in our material assembly process. In addition, compared to traditional porous scaffolds, our material not only possesses excellent biocompatibility and biodegradability but also benefits from a highly interconnected superporous structure that facilitates cell migration, tissue repair, and vascular regeneration.

Fig. S26. Cell viability test of LX-2 hepatic stellate cell. a) Fluorescence microscopy images of live-dead staining of LX-2 hepatic stellate cells after 1, 2, and 3 days of culture in A-spCS extract (n=3). b) The CCK-8 tests revealed the cell viability of LX-2 hepatic stellate cells cultured in different mass fractions (10, 20, 30, 40 mg/mL) of material extracts.

Fig. S30. A-spCS and GS in vivo degradation experiment. a) Schematic diagram of in vivo explanation. b) Representative photos and HE staining pictures of degradation of A-spCS and GS.

10. A large number of shape-memory hemostatic sponges have been reported, and their hemostatic effects have been verified in fatal arterial hemorrhage in large animals. The hemorrhage models have many options for effective hemostasis and are not challenges in first aid. The control hemostatic materials used do not rely on expansion to stop bleeding, so they cannot reflect the hemostatic advantages of the reported material more clearly. The authors need to compare the hemostasis advantages of this material with the reported shape-memory or expandable hemostatic materials.

Response:

We appreciate the reviewer's constructive comments and have made improvements to the study accordingly. We compared the hemostatic advantages of our material with shape memory or expandable hemostatic materials to more comprehensively assess our research findings.

Firstly, we supplemented the evaluation of the mechanical properties of commercial gelatin hemostatic sponges. Experimental results showed that the gelatin hemostatic sponge exhibited a stress of less than 20 kPa at 85% strain in compression testing, and its compression performance significantly decreased with increasing cycle numbers. Furthermore, in our rat liver puncture injury model hemostasis experiment (Fig. 6), the GS group showed greater blood lost (6.77 times greater than A-spCS) and longer hemostasis time (523% longer than A-spCS).

Shape memory sponges are expected to achieve rapid hemostasis by rapidly absorbing liquid and recovering shape, thereby constructing a physically robust barrier²⁴⁻²⁶. We additionally compared the shape recovery time of shape memory hemostatic sponges and hemostatic time of liver non-compressible wound model reported in the literature (Fig. S29), demonstrating that A-spCS exhibited a faster shape recovery ability for absorbing water/blood and more efficiently hemostasis.

Fig. S16. Stress-strain cyclic curves of gelatin sponges.

Hemostats	Xstat ³⁴	SHHS-CHS ³⁵	CMN-Cu ³⁶	TRAP/SP ³⁷	OBC-PDA/ PDA-MMT ₃ /Ag ³⁸	A-spCS
Shape-recovery time	25 s			10 s		4 s
Hemostatic time		55.50±7.00 s	47 s	33±2 s	32±4 s	12.67±12.5 s

Fig. S29. Comparison of shape-recovery time and hemostatic time of liver non-compressible wound model between the A-spCS and reported hemostats.

11. The manuscript also does not mention how much squeezing force the material can provide during expansion and hemostasis. This squeezing force also seems to be critical for hemostatic effect that relies on expansion properties.

Response:

We appreciate the reviewer's constructive comments. We supplemented the stress values at the stable state under compression to 85% strain and under the shape fixation state. We consider our material can provide stress exceeding 100 kPa when compressed to 85% strain for hemostatic purposes. At a shape fixation state (71.3% compression ratio), spCS can provide a

pressure of 32.8 kPa, which is significantly higher than commercial gelatin sponge (GS) (0.184 kPa). Although the stress decreases under deformation to the shape fixation state, in practical applications, we can mitigate the rebound tendency of the compressed material by using syringes or forceps to hold it in place. Therefore, the high stress provided by spCS, combined with its super anti-fatigue properties and elasticity, facilitates the rapid application of pressure to wounds and the establishment of hemostatic physical barriers.

Fig. S20. Compressive stress under different strain of gelatin sponge, CS, pCS, spCS, epCS. a) Compressive stress under 85% strain of CS, pCS, spCS, and epCS. b) Compressive stress under shape fixation ratio of CS, pCS, spCS, and epCS.

Reviewer #3:

The paper by Jiang et al. describes a new material for the treatment of non-compressible haemorrhages, which is an improvement of previously known material, chitosan. By using a specific pre-freezing temperature in the preparation of the material and coating its surface with hydrophobic dodecyl chains, the authors achieve improved pore size, better interconnection of the porous network, increased mechanical properties and improved coagulation ability of the material – which they name “superporous chitosan sponges” – spCS or “alkylated spCS” – A-spCS. The authors then explore the infiltration of spCS and A-spCS by red blood cells, platelets, coagulation and infection potential, and finally explore the material in the models of liver or liver/spleen injury in rats and mini-pigs, respectively. There is also a mention of scalability potential for the production of this material, which is essential if this material was to be used in the clinic.

The work constitutes a potential advance for the treatment of non-compressible haemorrhage.

However, there are several issues which authors should address before this paper can be published in Nature Communications.

Major comments:

1. In the investigation of strain-stress on the different materials, the authors describe better qualities of the spCS and A-spCS. Could the authors either investigate experimentally or comment on how the range described in the experiments correlates to actual values in human treated non-compressible haemorrhage?

Response:

We appreciate the reviewer's constructive comments. We believe the reviewer's feedback has prompted us to delve deeper into certain aspects of our research.

In our study, we conducted liver non-compressible hemorrhage control experiments on large animals (Bama miniature pigs), which is analogous to the situation of non-compressible hemorrhage control in humans. In our experiment, the material we provided accounts for approximately three-quarters of the wound size. When A-spCS is filled into the wound, its volume compression ratio should be $1 - (0.75 \times 0.75) = 43.75\%$. At this compression ratio, A-spCS can achieve effective and rapid hemostasis. According to our mechanical property experiment, spCS can provide a pressure of 123.7 kPa for the wound at a maximum compression ratio (>80%). At a shape fixation state (71.3% compression ratio), spCS can provide a pressure of 32.8 kPa (Fig. S20). Therefore, this implies that when a more effective hemostatic intervention is required, spCS can be further compressed, providing stress significantly higher than that achieved at the 43.75% compression ratio, thereby facilitating rapid hemostasis.

Fig. S20. Compressive stress under different strain of gelatin sponge, CS, pCS, spCS, epCS. a) Compressive stress under 85% strain of CS, pCS, spCS, and epCS. b) Compressive stress under shape fixation ratio of CS, pCS, spCS, and epCS.

2. In paragraph about pro-coagulation, the authors describe that both spCS and A-spCS are superior to other materials tested. While I agree with A-spCS being superior in the 3 methods tested, spCS only performed better in 1/3 tests used by the authors in Fig. 4 – the authors should tone down their statements here or provide additional data about spCS.

Response:

We appreciate the reviewer's constructive comments and have provided additional data for spCS.

In the vitro hemostasis experiments, both spCS and the control group consisted of unmodified chitosan, theoretically exhibiting no significant differences when blood volume is limited. Considering that actual non-compressible hemorrhage typically involves higher blood volumes, we modified the experimental conditions (100 mg material vs. 100 μ L blood) to increase the ratio of blood to material (100 mg material vs. 300 μ L blood) and conducted in vitro hemostasis experiments. The experimental results indicate that due to spCS's superior liquid absorption capacity, it demonstrates stronger in vitro hemostatic performance compared to the control group (Fig. S21). This suggests the potential of spCS as a promising hemostasis in scenarios characterized by higher blood loss, thereby highlighting its clinical relevance in managing non-compressible wound.

Fig. S21. Coagulation effect of various hemostatic materials and blood cell adhesion ability with. a) The Blood Clotting Index (BCI) of the hemostatic material tested at three times the normal blood volume for 10 minutes. b, c) Percentage of red blood cells and platelets adhered on different hemostatic materials at three times the normal blood volume.

3. In the test of anti-bacterial properties, could the authors comment on the E.coli and S.aureus infection prevalence after non-compressible haemorrhage? Are there any other bacterial infections known to be prevalent after this type of injury that could be relevant and tested here? Finally, could the authors comment on the 2h incubation with the bacteria in relation to real life situation, where the material might be exposed for longer to contamination?

Response:

We appreciate the reviewer's constructive comments and provide explanation for our choice and additional data for the concern.

The selection of Staphylococcus aureus (S. aureus) and Escherichia coli (E. coli) as research subjects in antibacterial experiments is multifaceted. Firstly, they are two common pathogenic bacteria with significant clinical relevance and research value. S. aureus and E. coli are prevalent human pathogens capable of causing various infections such as skin and soft tissue infections, urinary tract infections, and respiratory tract infections³⁹. They are major pathogens responsible for hospital-acquired and community-acquired infections, frequently encountered in clinical practice⁴⁰.

Secondly, S. aureus is a Gram-positive bacterium, while Escherichia coli is a Gram-negative bacterium, a distinction arising from differences in their cell wall structure⁴¹. Therefore, selecting these two bacteria for experimentation also allows for the exploration of materials'

antibacterial properties against Gram-positive and Gram-negative bacteria, providing a deeper understanding of the materials' antibacterial mechanisms. This investigation is of paramount importance for the development of new antibacterial materials and therapeutic approaches, offering theoretical support and practical guidance for the prevention and treatment of infectious diseases.

Additionally, we incorporated another strain of bacteria known to readily induce infection into our antimicrobial experiments. *Pseudomonas aeruginosa* (*P. aeruginosa*), is a common pathogenic bacterium known to cause various infections, particularly in immunocompromised individuals or healthcare settings⁴⁰. For instance, it can lead to skin and soft tissue infections, especially at sites of trauma or surgery. *P. aeruginosa* exhibits high levels of antibiotic resistance, complicating the treatment of infections. Our 2-hour co-incubation antibacterial assay demonstrated the effective antibacterial activity of our A-spCS group against *P. aeruginosa* (Fig. S23).

Fig. S23. a) Photographs of colonies of *P. aeruginosa* grown on LB agar plates after contact with blank, gauze, GS, CELOX, CELOX-E, CS, pCS, spCS, and A-spCS. b) Corresponding statistical results of colony counts of *P. aeruginosa*.

Regarding the issue of prolonged antibacterial efficacy, the pre-hospital time for civilian trauma generally varies due to regional and other factors. Typically, ambulance arrival times may range from 10 minutes to 1 hour. However, in the face of more complex battlefield environments, transportation times to medical facilities are often longer. Therefore, we conducted experiments to assess the antibacterial effectiveness of our material over extended durations of 4 and 6 hours. The experimental results demonstrate that our material maintains excellent antibacterial capability even after 6 hours, indicating its potential suitability for prolonged antibacterial applications (Fig. S24).

Fig. S24. a) Photographs of colonies of *E. coli*, *S. aureus*, and *P. aeruginosa* grown on LB agar plates after contact with blank and A-spCS for 4 h b) and 6 h. c) Corresponding statistical results of colony counts for 4 h d) and 6 h.

Furthermore, our superporous chitosan sponge, as a highly porous carrier rich in amino groups, can effectively enhance its long-term antibacterial capability by simply attracting or chemically grafting various antibacterial advantageous groups or molecules. For instance, we loaded antibiotics into the porous network of A-spCS, achieving sustained release of antibiotics and antibacterial results at 12, 24, and 48 hours indicate that the antibiotic-loaded sponge exhibits prolonged antibacterial efficacy (Fig. S25).

Fig. S25. The evaluation of the antimicrobial efficacy of penicillin G-loaded chitosan sponge for 12, 24, and 48 hours. a) Absorbance of LB after incubation of *E. coli*, *S. aureus*, and *P. aeruginosa* with penicillin-loaded chitosan sponge for 12, 24, and 48 hours. b) Penicillin G sustained release curve.

4. The authors should use another type of cells instead of 3T3 embryonic fibroblasts for testing compatibility of cells with their material, which is more relevant to the cells encountered in this type of damage. A good choice here would be primary liver or splenic cells, or at least liver/spleen cell line.

Response:

We are grateful for the reviewer's meticulous review, which has strengthened the rigor and validity of our findings.

Hepatic stellate cells (HSCs) play a crucial role in liver injury repair. Upon liver injury, HSCs become activated and transition into fibroblast-like cells, participating in the process of liver fibrosis. They synthesize extracellular matrix proteins such as collagen and secrete matrix metalloproteinases, contributing to the remodeling of liver matrix and modulation of the liver injury repair process^{42,43}. To better simulate the actual conditions in the liver, we conducted biocompatibility testing of the material using LX-2 hepatic stellate cells. Our live/dead staining assay and CCK-8 cell viability test revealed excellent cell compatibility of the material at different concentrations. Additionally, cell compatibility testing with commercial gelatin sponges showed no significant differences between our material and the gelatin sponge.

Fig. S26. Cell viability test of LX-2 hepatic stellate cell. a) Fluorescence microscopy images of live-dead staining of LX-2 hepatic stellate cells after 1, 2, and 3 days of culture in A-spCS extract (n=3). b) The CCK-8 tests revealed the cell viability of LX-2 hepatic stellate cells cultured in different mass fractions (10, 20, 30, 40 mg/mL) of material extracts.

5. The investigation of liver cell infiltration in Fig.8 is insufficient, and the following should be corrected:

- Could the authors better outline which part of the tissue is the materials tested and which is the liver? While I can guess, it would be better to specify this in the figure itself. As it is, I am not convinced that A-spCS is better than the other materials tested in this experiment.

Response:

We appreciate the reviewer's constructive comments. We have revised and added indications for the tissue and material sections in Fig. 8b, and replaced the images with higher-resolution ones. Based on counts and area measurements of the green fluorescent areas in the material region using ImageJ, statistical analysis revealed that A-spCS exhibits a higher degree of promotion in vascular and hepatic cell regeneration. This can be attributed to the highly porous interconnected superporous structure of A-spCS processed through the TA-2ND NC strategy, which is more conducive to cell migration and substance transport.

Fig. 8. b) H&E staining showing tissue ingrowth. DAPI staining showed that the host cells migrated into the interior of the sponge. Immunofluorescent staining for von Willebrand factor (VWF), albumin (ALB)-positive cells, and hepatocyte nuclear factor-4α (HNF-4α)

revealed the obvious infiltration of capillaries and liver parenchymal cells (LPC) within the A-spCS sponges, respectively. The dashed line represents the interface between material and tissue. Asterisks indicate the location of the implant sponges in vivo.

- the pictures in Fig.7b are of insufficient quality to assess the infiltration of different cells into the material. Could the authors improve the resolution? How exactly was the quantification performed?

Response:

We appreciate the reviewer's constructive comments and provide additional explanation for the concern.

We reacquired images using confocal microscopy and replaced them with higher-resolution images (Fig .8b) for statistical analysis. Image analysis was performed using ImageJ version 1.53k. For the area of cell migration into the material, we outlined the area below the dashed line as the total area (A_t) and selected the area below the dashed line of the H&E-stained section as the area of cell migration (A_m). The calculation formula is as follows:

$$\text{Cell migration area} = \frac{A_t}{A_m} \times 100\%$$

For immunofluorescence quantification, we adjusted the threshold to select the fluorescence intensity of blue or green in the whole picture of the blank control group, denoted as F_b , and the total fluorescence intensity of blue or green below the dashed line in the material group, denoted as F_m . The calculation formula is as follows:

$$\text{Cell migration area} = \frac{F_t}{F_m} \times 100\%$$

Fig. 8. b) H&E staining showing tissue ingrowth. DAPI staining showed that the host cells migrated into the interior of the sponge. Immunofluorescent staining for von Willebrand factor (VWF), albumin (ALB)-positive cells, and hepatocyte nuclear factor-4 α (HNF-4 α) revealed the obvious infiltration of capillaries and liver parenchymal cells (LPC) within the A-spCS sponges, respectively. The dashed line represents the interface between material and tissue. Asterisks indicate the location of the implant sponges in vivo.

- HNF4a is not a cytokine, but a nuclear transcription factor – please correct lines 415-416 and the whole sentence for accuracy.

Response:

We appreciate the reviewer's detailed scrutiny of our work and we have addressed the issues raised in the revised version.

Revised version:

Liver development was also evaluated by immunostaining for hepatocyte nuclear factor-4 α (HNF-4 α), which is a nuclear transcription factor associated with the generation of hepatocyte, which indicated that the A-spCS group contained a much higher liver cell number

than CELOX, CS, and pCS groups.

- Finally, Fig.8g should be performed with liver-relevant cells, and crucially - also comparing migration to other materials tested in the paper, and not shown in isolation as it is now. If authors want to show relevance for the migratory ability of liver fibroblast cells, the use of primary cells or at least hepatic stellate cell line should be employed here.

Response:

We appreciate the reviewer's constructive comments and provide additional data for the concern.

Therefore, we selected the CS and pCS groups as controls for in vitro cell migration. In comparison with the control group, LX-2 hepatic stellate cells appeared and spread in the middle and lower regions of the A-spCS group, confirming the promotion of cell infiltration and growth by A-spCS (Fig. S31). Additionally, no LX-2 cells were observed at the bottom of the CS group, possibly due to the restricted pore connectivity of CS, which hindered cell migration to the bottom region.

Fig. S31. In vitro cell migration experiments of LX-2 within CS, pCS, and A-spCS.

Minor comments:

- For the live/dead staining, authors only show Calcein live stain. Was any dead stain used? How were the cells counted for the assessment of viability? If no dead cell stain was used, the authors should correct this in the text.

Response:

We appreciate the reviewer's constructive comments. For live/dead staining, we used propidium iodide as a dead stain in our live/dead staining experiments. The likely reason for not observing red fluorescence could be due to the removal of suspended dead cells during the dye washing process. Therefore, we adjusted the experimental procedure and repeated the experiment. Upon direct observation after dye addition and incubation, we observed red fluorescence (Fig. 5f). Moreover, we conducted experiments with LX-2 cells which also indicated favorable biocompatibility of A-spCS (Fig. S26a). Regarding quantitative aspects, our focus within the viability/dead staining experiments primarily centered on qualitatively assessing the cellular vitality and growth status. Quantitative determination of cell viability was predominantly achieved through the MTT/CCK-8 assay (Fig. 5g and Fig. S26b). This assay provided quantitative data regarding cell viability, complementing the qualitative observations obtained through viability/dead staining.

Fig. 5. f) Fluorescence microscopy images of live-dead staining of 3T3 fibroblasts after 1, 2, and 3 days of culture in A-spCS extract. g) The MTT tests revealed the cell viability of 3T3 fibroblasts cultured in different mass fractions (10, 20, 30, 40 mg/mL) of A-spCS extracts.

Fig. S26. a) Fluorescence microscopy images of live-dead staining of LX-2 hepatic stellate cells after 1, 2, and 3 days of culture in A-spCS extract ($n=3$). b) The CCK-8 tests revealed the cell viability of LX-2 hepatic stellate cells cultured in different mass fractions (10, 20, 30, 40 mg/mL) of material extracts.

- Could the authors please include a title in each of their supplementary videos? It is very hard to follow the different materials and techniques as it is, without any title embedded within the videos.

Response:

We appreciate the reviewer's constructive comments. We have embedded titles in each supplemental video for reference purposes, which may facilitate viewers' understanding and navigation of the video content.

- In figure 3, panels k, l and m are very hard to decipher; perhaps use of different colours could be implemented by the authors here to improve clarity or simply a bigger panel with higher resolution?

Response:

We appreciate the reviewer's constructive comments. We enhanced the clarity of Figure 3k, l, and m by employing more vibrant colors to improve visual resolution. Additionally, we further increased the image resolution.

Fig. 3. *k, l, m*) Stress-strain cyclic curves of CS, pCS, and spCS (“*k*” represents the hysteresis in the first cycle of each group).

Reference

- Ji, D. & Kim, J. Recent strategies for strengthening and stiffening tough hydrogels. *Advanced NanoBiomed Research* **1**, 2100026 (2021).
- Hua, M. *et al.* Strong tough hydrogels via the synergy of freeze-casting and salting out. *Nature* **590**, 594-599 (2021).
- Huang, S. *et al.* Control of polymers’ amorphous-crystalline transition enables miniaturization and multifunctional integration for hydrogel bioelectronics. *Nature Communications* **15**, 3525 (2024).
- Zhu, S. *et al.* Bioinspired structural hydrogels with highly ordered hierarchical orientations by flow-induced alignment of nanofibrils. *Nature Communications* **15**, 118 (2024).
- Alsaid, Y. *et al.* Tunable sponge-like hierarchically porous hydrogels with simultaneously enhanced diffusivity and mechanical properties. *Advanced materials* **33**, 2008235 (2021).
- Mredha, M. T. I. *et al.* A facile method to fabricate anisotropic hydrogels with perfectly aligned hierarchical fibrous structures. *Advanced Materials* **30**, 1704937 (2018).
- Sun, X., Mao, Y., Yu, Z., Yang, P. & Jiang, F. A Biomimetic “Salting Out–Alignment–Locking” Tactic to Design Strong and Tough Hydrogel. *Advanced Materials*, 2400084 (2024).
- Kim, J., Zhang, G., Shi, M. & Suo, Z. Fracture, fatigue, and friction of polymers in which entanglements greatly outnumber cross-links. *Science* **374**, 212-216 (2021).
- Lan, L. *et al.* Skin-Inspired All-Natural Biogel for Bioadhesive Interface. *Advanced Materials*, 2401151 (2024).
- Reys, L. L. *et al.* Influence of freezing temperature and deacetylation degree on the performance of freeze-dried chitosan scaffolds towards cartilage tissue engineering. *European Polymer Journal* **95**, 232-240 (2017).
- Takeshita, S., Zhao, S., Malfait, W. J. & Koebel, M. M. Chemistry of chitosan aerogels: three-dimensional pore control for tailored applications. *Angewandte Chemie International Edition* **60**, 9828-9851 (2021).
- Wang, M. *et al.* Hierarchical porous chitosan sponges as robust and recyclable adsorbents for anionic dye adsorption. *Scientific reports* **7**, 18054 (2017).
- Song, W. *et al.* Preparation of freeze-dried porous chitosan microspheres for the removal of hexavalent chromium. *Applied Sciences* **11**, 4217 (2021).
- He, H. *et al.* Efficient, biosafe and tissue adhesive hemostatic cotton gauze with controlled balance of hydrophilicity and hydrophobicity. *Nature Communications* **13**, 552 (2022).
- Dowling, M. B. *et al.* A self-assembling hydrophobically modified chitosan capable of reversible hemostatic

action. *Biomaterials* **32**, 3351-3357 (2011).

16. Chen, G. *et al.* Bioinspired multifunctional hybrid hydrogel promotes wound healing. *Advanced Functional Materials* **28**, 1801386 (2018).

17. Du, X. *et al.* Anti-infective and pro-coagulant chitosan-based hydrogel tissue adhesive for sutureless wound closure. *Biomacromolecules* **21**, 1243-1253 (2020).

18. Qi, L. *et al.* Bioinspired multiscale micro-/nanofiber network design enabling extremely compressible, fatigue-resistant, and rapidly shape-recoverable cryogels. *ACS nano* **17**, 6317-6329 (2023).

19. Ye, X., Bishop, A. M., Reidy, J. A., Needham, L. L. & Calafat, A. M. Parabens as urinary biomarkers of exposure in humans. *Environmental health perspectives* **114**, 1843-1846 (2006).

20. Ye, X., Wong, L.-Y., Jia, L. T., Needham, L. L. & Calafat, A. M. Stability of the conjugated species of environmental phenols and parabens in human serum. *Environment international* **35**, 1160-1163 (2009).

21. Balouiri, M., Sadiki, M. & Ibsouda, S. K. Methods for in vitro evaluating antimicrobial activity: A review. *Journal of pharmaceutical analysis* **6**, 71-79 (2016).

22. Serra, R. *et al.* Chronic wound infections: the role of *Pseudomonas aeruginosa* and *Staphylococcus aureus*. *Expert review of anti-infective therapy* **13**, 605-613 (2015).

23. Kim, M. *et al.* Analysis of shape memory behavior and mechanical properties of shape memory polymer composites using thermal conductive fillers. *Micromachines* **12**, 1107 (2021).

24. Lv, S., Cai, M., Leng, F. & Jiang, X. Biodegradable carboxymethyl chitin-based hemostatic sponges with high strength and shape memory for non-compressible hemorrhage. *Carbohydrate Polymers* **288**, 119369 (2022).

25. Zheng, W. *et al.* A novel pullulan oxidation approach to preparing a shape memory sponge with rapid reaction capability for massive hemorrhage. *Chemical Engineering Journal* **447**, 137482 (2022).

26. Zhao, X., Guo, B., Wu, H., Liang, Y. & Ma, P. X. Injectable antibacterial conductive nanocomposite cryogels with rapid shape recovery for noncompressible hemorrhage and wound healing. *Nature communications* **9**, 2784 (2018).

27. Levine, S., Lowndes, J., Watson, E. J. & Neale, G. A theory of capillary rise of a liquid in a vertical cylindrical tube and in a parallel-plate channel: Washburn equation modified to account for the meniscus with slippage at the contact line. *Journal of Colloid and Interface Science* **73**, 136-151 (1980).

28. Xi, G. *et al.* Polysaccharide-based lotus seedpod surface-like porous microsphere with precise and controllable micromorphology for ultrarapid hemostasis. *ACS applied materials & interfaces* **11**, 46558-46571 (2019).

29. Li, N. *et al.* A Natural Self-Assembled Gel-Sponge with Hierarchical Porous Structure for Rapid Hemostasis and Antibacterial. *Advanced Healthcare Materials* **12**, 2301465 (2023).

30. Lerouge, T., Pitois, O., Grande, D., Le Droumaguet, B. & Coussot, P. Synergistic actions of mixed small and large pores for capillary absorption through biporous polymeric materials. *Soft Matter* **14**, 8137-8146 (2018).

31. Landsman, T. *et al.* A shape memory foam composite with enhanced fluid uptake and bactericidal properties as a hemostatic agent. *Acta biomaterialia* **47**, 91-99 (2017).

32. Nagasue, N., Yukaya, H., Ogawa, Y., Kohno, H. & Nakamura, T. Human liver regeneration after major hepatic resection. A study of normal liver and livers with chronic hepatitis and cirrhosis. *Annals of surgery* **206**, 30 (1987).

33. Kim, D.-H. *et al.* Bioengineered liver crosslinked with nano-graphene oxide enables efficient liver regeneration via MMP suppression and immunomodulation. *Nature Communications* **14**, 801 (2023).

34. Mueller, G. R. *et al.* A novel sponge-based wound stasis dressing to treat lethal noncompressible hemorrhage. *Journal of Trauma and Acute Care Surgery* **73**, S134-S139 (2012).

35. Hu, Z. *et al.* A super hydrophilic and high strength chitosan hemostatic sponge prepared by freeze-drying and

alkali treatment for rapid hemostasis. *Materials Today Communications*, 108855 (2024).

36. Huang, Y. *et al.* MXene-NH₂/chitosan hemostatic sponges for rapid wound healing. *International Journal of Biological Macromolecules* **260**, 129489 (2024).

37. Yang, X. *et al.* Peptide-immobilized starch/PEG sponge with rapid shape recovery and dual-function for both uncontrolled and noncompressible hemorrhage. *Acta Biomaterialia* **99**, 220-235 (2019).

38. Cao, S., Yang, Y., Zhang, S., Liu, K. & Chen, J. Multifunctional dopamine modification of green antibacterial hemostatic sponge. *Materials Science and Engineering: C* **127**, 112227 (2021).

39. Worlicek, M. *et al.* Splanchnic sympathectomy prevents translocation and spreading of E coli but not S aureus in liver cirrhosis. *Gut* **59**, 1127-1134 (2010).

40. Bessa, L. J., Fazii, P., Di Giulio, M. & Cellini, L. Bacterial isolates from infected wounds and their antibiotic susceptibility pattern: some remarks about wound infection. *International wound journal* **12**, 47-52 (2015).

41. Yasuyuki, M. *et al.* Antibacterial properties of nine pure metals: a laboratory study using Staphylococcus aureus and Escherichia coli. *Biofouling* **26**, 851-858 (2010).

42. Ezhilarasan, D. *et al.* Silibinin inhibits proliferation and migration of human hepatic stellate LX-2 cells. *Journal of Clinical and Experimental Hepatology* **6**, 167-174 (2016).

43. Iredale, J. P. in *Seminars in liver disease*. 427-436 (Copyright© 2001 by Thieme Medical Publishers, Inc., 333 Seventh Avenue, New ...).

REVIEWERS' COMMENTS

Reviewer #2 (Remarks to the Author):

The revised work can be accepted in its current form.

Reviewer #3 (Remarks to the Author):

I congratulate the authors for their changes to the paper, especially for the new data strengthening the spCS comparisons, extended data on the bacterial tests – now very convincing (including new timepoints and an additional bacterial species, as well as in-text explanations), and including liver-relevant hepatic stellate cells for the assessment of infiltration capacity. I am happy to recommend this revised manuscript for publication.